# The global impact of bacterial processes on carbon mass

**Barbara Ervens and Pierre Amato**

Université Clermont Auvergne, CNRS, Sigma-Clermont, Institut de Chimie de Clermont-Ferrand, 63000 Clermont-Ferrand, France

*Correspondence to:* barbara.ervens@uca.fr and pierre.amato@uca.fr

**Abstract.** Many recent studies have identified biological material as a major fraction of ambient aerosol loading. A
small fraction of these bioaerosols consist of bacteria that have attracted a lot of attention due to their role in cloud formation and adverse health effects. Current atmospheric models consider bacteria as inert quantities and neglect cell growth and multiplication. We provide here a framework to estimate the production of secondary biological aerosol (SBA) mass in clouds by microbial cell growth and multiplication. The best estimate of SBA formation rates of 3.7 Tg $yr^{-1}$ is comparable to previous model estimates of the primary emission of bacteria into the atmosphere, and thus might
represent a previously unrecognized source of biological aerosol material. We discuss in detail the large uncertainties associated with our estimates based on the rather sparse available data on bacteria abundance, growth conditions and properties. Additionally, the loss of water-soluble organic carbon (WSOC) due to microbial processes in cloud droplets has been suggested to compete under some conditions with WSOC loss by chemical (OH) reactions. Our estimates suggest that microbial and chemical processes might lead to a global loss of WSOC of 8 - 11 Tg $yr^{-1}$ and 8 - 20 Tg $yr^{-1}$
, respectively. While also this estimate is very approximate, the analysis of the uncertainties and ranges of all parameters suggests that high concentrations of metabolically active bacteria in clouds might represent an efficient sink for organics. Our estimates also highlight the urgent needs for more data concerning microbial concentrations, fluxes and activity in the atmosphere to evaluate the role of bacterial processes as net aerosol sink or source on various spatial and temporal scales.

## 1. Introduction

The characterization and quantification of outdoor bioaerosols is an active field of current atmospheric research since bioaerosols have been suggested to contribute to adverse health effects and cloud formation as ice-nucleating particles (Després et al., 2012). Biological material includes debris, pollen, bacteria, fungal spores, and viruses and is usually considered as being directly emitted to the atmosphere (primary biological aerosol, PBA (Jaenicke, 2005)). The total
number and mass concentrations of PBA particles vary widely in space and time: Posfai et al. (1998) found 1% of particles with biological material above the Southern Ocean whereas Artaxo et al. (1990) identified more than 90% of all particles to contain biological material during the wet season in the Amazon. In an urban/remote region in Germany, 24% of all particles were found to include a biological fraction (Matthias-Maser and Jaenicke, 2000). Similar

concentrations were observed at a remote high-altitude site with 16 - 64% of the mass of particles with diameters of less than 10 μm being composed of biological mass (Wiedinmyer et al., 2009) whereas the PBA number fraction was much smaller (0.3 - 18%) in Rome, Italy (Perrino and Marcovecchio, 2016). Bacteria only comprise a small fraction of the total biological aerosol mass but they alone can contribute up to about ~20% of the total number of particles with diameters greater than 0.5 μm (Bowers et al., 2012).

Near the ground, typical concentrations of total airborne bacteria range from ~$10^2$ to $10^6$ cells m$^{-3}$, depending on the emission source (Burrows et al., 2009b), and on temporal, meteorological, and other environmental conditions influencing its propensity to emit particles to the air (Carotenuto et al., 2017; Huffman et al., 2013; Lighthart, 1997; Lighthart and Shaffer, 1995). Atmospheric mixing aloft tends to homogenize the number and diversity of the various bacteria types as the distance from sources increases. In the free troposphere, concentrations of ~10,000 cells m$^{-3}$ are reported, including in clouds (DeLeon-Rodriguez et al., 2013; Vaïtilingom et al., 2013). Some extent of selection toward certain species of bacteria probably occurs during aerosolization and atmospheric transport (Joly et al., 2015; Michaud et al., 2018). However, such selection has not been clearly proven yet as the bacterial assemblages found at high altitude often resemble those observed near the ground (Amato et al., 2017; DeLeon-Rodriguez et al., 2013; Smith et al., 2018).

The atmosphere is a harsh environment for living microorganisms: low temperatures at high altitude, UV radiation (Madronich et al., 2018) and high free radical concentrations (Haddrell and Thomas, 2017; Marinoni et al., 2011) are thought to greatly challenge living organisms (Amato et al., 2019; Joly et al., 2015; Smith et al., 2011). Additionally, the rapidly changing conditions in clouds, like condensation/evaporation and freeze/thaw cycles, can cause strong physiological shocks and physical damages to cells, which can eventually be lethal. The viability of airborne microorganisms is thus very variable in space and time depending on environmental conditions (Fahlgren et al., 2010; Hu et al., 2017; Lighthart and Shaffer, 1995; Monteil et al., 2014), but yet the fact that a fraction of bacteria cells are viable was shown in many experiments of microbiological cultures from ambient aerosol samples (Amato et al., 2007b; Bovallius et al., 1978; Lighthart, 1997; Newman, 1948). This was specified and quantified more recently by direct observations and measurements of biological activity imprints (Amato et al., 2007a, 2017; Sattler et al., 2001; Wirgot et al., 2017). The multiplication of airborne bacteria was observed from aerosols generated from bacteria cultures (Dimmick et al., 1979), as well as in natural polluted fog (Fuzzi et al., 1997). Thus, the estimated PBA emissions might be biased high as bacteria cell growth and multiplication provides an additional source of bacteria mass and, thus, observed bacteria concentrations represent the sum of emission fluxes that are smaller than assumed and the secondary production in the atmosphere.

Efficient bacteria cell growth and multiplication are largely constrained by the presence of liquid water (Davey, 1989; Haddrell and Thomas, 2017). One can thus assume that microbial processes in the atmosphere are limited to the time microorganisms spend in clouds (*Figure 1*). Cell growth and multiplication lead to an increase of the initial cell mass and to more biological material (Kaprelyants and Kell, 1993; Norris, 2015; Si et al., 2017) whereas bacteria dormancy and death do not lead to any change in cell mass (Engelberg-Kulka et al., 2006; Kaprelyants and Kell, 1993; Price and Sowers, 2004). We introduce the term 'secondary biological aerosol' (SBA) mass here in order to distinguish this

aerosol source from directly emitted PBA. Heterotrophic bacterial processes require the uptake of organic substrates by the cells, which are subsequently converted by metabolic processes into new organic products, biochemical energy and $CO_2$ ('respiration', *Figure 1*). These substrates include organics (e.g., carboxylic acids, sugars); other elements (e.g., nitrogen, phosphorous, potassium, metals) are also needed and exist in bioavailable forms in cloud water. The

biotransformation of formate, acetate, succinate, lactate, oxalate, and formaldehyde (Ariya et al., 2002; Vaïtilingom et al., 2010), phenol (Lallement et al., 2018) and methane (Šantl-Temkiv et al., 2013) by bacteria and fungi was studied in aqueous solution mimicking the typical chemical composition of cloud water, and it was suggested that under specific conditions, microbial processes might be competitive to chemical radical processes as sinks for these compounds (Delort et al., 2010; Vaïtilingom et al., 2011, 2013). The efficiency of such metabolic processes strongly

depends on the bacteria types, substrates and their availability within the cloud droplets. In the present study, we perform an estimate of the global importance of SBA formation and microbial WSOC loss. All parameters and their uncertainties are discussed based on the sparse data sets currently available.

## 2.   Data and assumptions on bacterial processes in clouds

### 2.1 Atmospheric concentrations of bacteria cells

Burrows et al. (2009a, 2009b) have summarized data on number concentrations and emission fluxes of bacteria above various ecosystems on the Earth surface. These ecosystems represent lumped categories based on the original classification by Olson et al. (1992). The compilation by Burrows et al. (2009) also includes the estimates of cell concentrations near the surface in a range of 10,000 $m^{-3}$ (seas) up to 650,000 $m^{-3}$ (urban). The category 'seas' in *Table 1* is not included in the original categories as defined by Olson (1992). However, it was added by Burrows et al (2009b)

in order to represent a full global coverage. There are only a few studies available that report measurements of bacteria numbers in the air above the Arctic. For example, Šantl-Temkiv et al (2018) report $(1.3 \pm 1.0) \cdot 10^3$ cells $m^{-3}$ above partially glaciated surfaces in Southwest Greenland. Recent large scale microbiological studies including those from a number of ground based stations around the globe (Dommergue et al., 2019; Tignat-Perrier et al., 2019) reported bacteria abundances over north Greenland in the Arctic (Station Nord) three to four orders of magnitude lower than

anywhere elsewhere on the planet with the exception of Antarctica, with 16S rRNA gene copy numbers (representing the uppermost expectable cell concentration) of $(7.3 \pm 9.2) \cdot 10^2 \, m^{-3}$. In this area the air content is affected by emissions from sea ice, the Arctic ocean and long-range transport from northern Eurasia. This number is even lower than the one estimated above land ice by Burrows et al. (2009b) (5000 $m^{-3}$). Thus, it can be assumed that the contribution of bacteria to SBA formation above sea ice is negligible.  Bacteria populations aloft represent a mixture of bacteria that were

emitted from different ecosystems and subsequently mixed (Burrows et al., 2009b). Despite these mixing processes, there are bacteria types that can be considered characteristic for each ecosystem (Wéry et al., 2017). *Table 1* lists cell concentrations as published by Burrows et al. (2009b) complemented by some more recent measurements. We extend this overview by data on bacteria types, suggested as predominant or characteristic for each ecosystem. In several cases, more than one predominant bacteria type is listed as specific geographical, meteorological and other

environmental conditions might lead to differences in the diversity of bacteria populations for the same category of ecosystem. We also provide global average data (Category 'All') and define one of the most abundant bacteria type

(Alpha-Proteobacteria) alive in the atmosphere (Amato et al., 2019; Klein et al., 2016) as a representative type. Several studies report total concentrations of bacteria cells in the atmosphere whereas others present only the concentration of viable cells. The complexity of distinguishing viable, cultivable and dead bacteria cells in the atmosphere has been discussed in several studies (Burrows et al., 2009a; Otero Fernandez et al., 2019).

We assume in *Section 3* that all bacteria cells as listed in *Table 1* are metabolically active. The atmospheric lifetime of bacteria cells is limited to several minutes (Otero Fernandez et al., 2019) to hours (Amato et al., 2015). In our estimates, we neither include assumptions on the limited lifetime of bacteria cells nor on their residence time in the atmosphere as it is assumed that PBA emissions lead to a continuous replenishment of bacteria in the atmosphere resulting in a steady-state concentration of living cells. The consequences of limited cell life- and residence time on SBA formation warrant further studies in more sophisticated model approaches.

**2.2 Cell generation rates $R_{Cell}$**

Different levels of metabolic activity can be distinguished, from survival, where cells only repair molecular damages, to maintenance (dormancy), where cells do not divide but maintain biological functions, to growth, allowing the net production of biological mass (Price and Sowers, 2004) (*Figure 1*). The generation rate of a microorganism during growth is probably the most common microbiological criterion used for characterizing microbial multiplication in the laboratory; it corresponds to the time that is needed for doubling the cell number, i.e., for producing two "children" cells from one individual. This requires mass production from nutrients that provide the necessary molecular bricks and biochemical energy. The activity depends on physiological traits of the microorganism, with optima at a given temperature, pH, salinity, and other conditions that define its fitness for its habitat. The generation time of bacteria at their optimum growth conditions usually ranges from ~20 minutes (Marr, 1991) to several days or weeks; as conditions deviate from the optima, this lengthen to virtually infinite time in non-dividing cells.

The *cellular* growth rate itself, i.e. the increase of individual cell size and mass, is intimately linked with generation time: cell size increases in a predictable way as generation time decreases (Si et al., 2017), and it can vary by a factor of up to eight within a single bacteria species. Compared to generation rates, cellular growth rates are usually small and, thus, in the following only data for generation rates are used to estimate SBA mass formation rates.

Metabolic activity, in terms of carbon uptake per units of biomass and time, can range over more than ten orders of magnitude, depending on many factors of which temperature is a major one (Price and Sowers, 2004). Therefore, if available, temperature-dependent generation rates $R_{cell}$ are listed in *Table 1* and shown in *Figure 2*. In addition, the highest expectable growth rate for bacteria as measured under laboratory conditions in culture medium is also shown in the figure for constraining an upper theoretical limit. This corresponds to the generation rate of the laboratory model *Escherichia coli* under optimal conditions (37°C). However, it can be expected that this is not representative of situations encountered in clouds. Generally, the temperature dependence of cell generation rates can be scaled by the empirical relationship in Eq-1:

$$R_{cell}(T_2) = R_{cell}(T_1) \cdot Q_{10}^{(T_2 - T_1)/10} \tag{1}$$

whereas $R(T_1)$ and $R(T_2)$ are the generation rates ($h^{-1}$) at two temperatures $T_1$ and $T_2$. $Q_{10}$ is a dimensionless scaling factor that expresses the change of these rates over an interval of 10°C and typically has values between two and three within relatively small temperature intervals (Lipson et al., 2002; Sand-Jensen et al., 2007). In general, Equation (1) can be applied for all bacteria types and is usually valid for generation rates in liquid water over a temperature range up to ~25°C; however, the slope ($Q_{10}$ factor) and the maximum temperature depend on the bacteria type. In *Figure 2*, the dashed lines towards lower temperatures represent extrapolations of the generation rates at ~20°C reflecting the general agreement between measured and calculated temperature dependencies using $Q_{10}$ = 2 or 3, respectively. Using generation rates measured at ~20°C might lead to an overestimate for SBA mass formation rates in colder clouds. However, we chose these values for the calculations in Section 3 as most experimentally-derived growth rates are available for temperatures of ~ 20 – 30°C. Generally, at temperatures below 0°C, cell metabolic activity is negligible in terms of carbon flux even though cells can maintain and survive under such conditions (Amato et al., 2009, 2010; Price and Sowers, 2004).

## 2.2 Bacteria growth efficiency (BGE)

Chemoheterotrophs - representatives of which were shown to maintain metabolic activity in clouds (Amato et al., 2017) - take up carbon from dissolved organic material for both recovering biochemical energy and converting the substrates into $CO_2$ and other products. Bacteria growth efficiency (BGE) is defined as the biological mass that is produced relatively to the amount of carbon taken up from the environment, the rest being converted into $CO_2$ (Eiler et al., 2003):

$$BGE = \frac{d[\text{organic products}]}{d[\text{organic products}] + d[CO2]} \tag{2}$$

Note that in the original literature BGE is defined as a measure of 'biomass production' instead of 'organic products' in *Eq (2)*. Given the large body of atmospheric literature on aerosol processes that discusses 'biomass' as material from any living matter (e.g., aerosol from forest fires), we avoid using 'biomass' in the current context of microbial processes. BGE for planktonic bacteria range from < 0.4% to 80% with the highest values for eutrophic conditions (Eiler et al., 2003). In turn, in the same study, it was shown that, when substrate availability is limited, values from ~7% to ~14% are generally observed. As the conditions in cloud water can be considered oligotrophic with typical concentrations of dissolved organic carbon (DOC) of less than 0.1 mM (Herckes et al., 2013), low BGEs in the range of 0.1 – 10% can be expected, i.e. DOC is efficiently converted into $CO_2$. Bacteria cells are composed not only of carbon, but also other elements such as nitrogen, and phosphorus, the proportions of which can vary widely depending on nutrient condition (e.g., Vrede et al., 2002; Chrzanowski and Kyle, 1996). Hence, the total biological mass produced during cell growth and multiplication is higher than the amount of DOC incorporated.

**2.3 Cloud properties relevant for microbial activity**

**2.3.1 Cell concentrations in cloud water**

Bacteria cells have sizes of up to several micrometers which explains their high efficiency to act as cloud condensation nuclei (CCN) (Bauer et al., 2003; Després et al., 2012). Assessing the hygroscopicity of biological particles is complex since it cannot be calculated in a similar way as for chemical compounds where total hygroscopicity represents the sum of the contributions of all components (Ariya et al., 2009). Once particles form cloud droplets, chemical compounds dissolving in cloud water will trigger the growth of the processed CCN and enhance hygroscopicity and CCN activity of aged particles in subsequent cloud cycles. The dissolution of ambient cell populations of 100 to 50,000 $m^{-3}$ (***Table 1***) results in 200 to 500,000 cells $mL^{-1}$ for clouds with liquid water contents (LWC) of 0.5 g $m^{-3}$ and 0.1 g $m^{-3}$, respectively. The reasonable agreement of cell concentrations outside of clouds and those in cloud water suggests that a large fraction of bacteria cells are scavenged and act as CCN.

Some bacteria are well-known to efficiently act as ice nuclei (Amato et al., 2015; Möhler et al., 2008; Morris et al., 2004). In the current study, we neglect the potential role of ice clouds as media of microbial metabolic activity. In addition to low temperatures resulting in very low generation rates (***Figure 2***), the substrate diffusion to the bacteria will be limited resulting in negligible consumption of dissolved carbon.

**2.3.2 Time fractions of microbial processes in clouds ($F_{cloud}$)**

As we assume that both SBA mass production and WSOC loss only occur when bacteria are suspended in cloud droplets, we need to estimate the time bacteria spend in liquid clouds. In general, cloud contact times, i.e., the time air spends in a cloud, are dependent on cloud depth and vertical velocity (Feingold et al., 2013). This small-scale information is not consistently available for the large regions as covered by the ecosystems listed in ***Table 1***. In order to give an estimate of the cloud processing time over the various large ecosystems as identified by Olson et al. (1992), we derived visually the cloud fractions during spring averaged for 2000 – 2011 from MODIS Terra (e.g., based on Figure 2b by King et al. (2013)). This visual approach to derive cloud fractions from the average maps neglects details on the variability of cloud fractions among the same ecosystem category in different geographic regions. For such categories (e.g. forests), the approximate surface contributions of the various regions were taken into account and averaged. fractions of the different regions. This representation gives only a general view of cloudiness that varies over smaller spatial and temporal scales. However, given the conceptual nature of our study that builds upon the lumped ecosystem categories as used in the previous study for primary bacteria emissions by Burrows et al. (2009b), our approach seems appropriate to give (i) an order-of-magnitude estimate of cloudiness above the various ecosystems and (ii) enough detail of its concept to be refined in future studies on smaller spatial and temporal scales. Globally, a range of cloud thicknesses of 1.4 – 1.9 km has been derived (Table 1 in (Wang et al., 2000)), from which we use 1.5 km as a single value for the average cloud thickness. Assuming further that globally >90% of all liquid clouds reside in the lowest 6 km of the atmosphere ($\Delta z$ = 6 km) (Pruppacher and Jaenicke, 1995), we can convert the cloud coverage as obtained from satellite data ($F_{clc}$ in ***Table 2***) into cloud volume fractions using Eq-3:

$$F_{cloud} = F_{clc} \cdot \frac{cloud\ thickness\ [km]}{\Delta z\ [km]} = F_{clc} \cdot \frac{1.5\ km}{6\ km} \qquad (3)$$

Comparison of previous global estimates of cloud coverage of 60% (Pruppacher and Jaenicke, 1995) and the volume fraction of liquid clouds within the atmosphere of 15% (Lelieveld and Crutzen, 1990) generally supports this relationship. The resulting $F_{cloud}$ values are summarized in **Table 2** together with the cloud coverage data ($F_{clc}$) and the percentage area fraction of each ecosystem of the Earth surface, taken from Burrows et al. (2009b) and originally obtained from Olson (1992).

## 3. Results and Discussion

### 3.1 SBA mass production

### 3.1.1 Calculation of SBA formation rates

We calculate the SBA mass formation rate [ng m$^{-3}$ day$^{-1}$] above each ecosystem $i$, using Eq (4):

$$\left(\frac{dm}{dt}\right)_{SBA,i,day} = R_{cell,i} \cdot F_{live} \cdot C_{cell,i} \cdot F_{cloud,i} \cdot m_{cell} \qquad (4)$$

where $R_{cell}$ is the cell generation rate [h$^{-1}$] (**Table 1**). For ecosystems, for which $R_{cell}$ of the representative bacteria types is not available, we assume the average formation rate of $R_{cell} = 0.3$ h$^{-1}$ as an upper limit for atmospherically-relevant conditions, corresponding to a generation time of approximately three hours. $C_{cell}$ denotes the ambient cell concentration [cell m$^{-3}$] (**Table 1**), $F_{live}$ is the fraction of living cells in total cell concentration and assumed to be unity here, $F_{cloud}$ is the fraction of total time when bacteria are active in clouds (**Table 2**), and $m_{cell}$ is the average mass of a single cell, independent of the bacteria type. The cell mass $m_{cell}$ is assumed to be $52 \cdot 10^{-15}$ g cell$^{-1}$ (Sattler et al., 2001), equivalent to a spherical particle with diameter of 500 nm and a density of 1 g cm$^{-3}$.

For nearly all ecosystems, predicted SBA formation rates are in the range of ~0.1 to ~1 ng m$^{-3}$ s$^{-1}$ (**Figure 3a**), with higher values for crops and shrubs where $C_{cell}$ were found to be highest (**Table 1**). The average value (0.6 ng m$^{-3}$ day$^{-1}$), calculated using the average values representative for all ecosystems (Category 'All' in **Table 1**), is similar to most of the formation rates in the individual ecosystems, suggesting that using these average data for a global estimate results in a reasonable order of magnitude of SBA formation. Only above land ice, where $C_{cell}$ is small, the rate is significantly smaller. Given that the temperatures above land ice might be on average lower than above other regions, the relative importance of SBA formation there might be even smaller. According to the definition of the categories as suggested by Olson (1992), the category 'land ice' does not include sea ice. It can be expected that above sea ice the sources and metabolic activity of bacteria are also very low (Martin et al., 2009) and thus can be likely neglected on a global scale.

To compare the mass production to other global aerosol sources, $m_{SBA,day}$ is converted into a production flux [Tg yr$^{-1}$] for each ecosystem $i$ and scaled by the surface fraction of each ecosystem:

$$P_{SBA,i} = \left(\frac{dm}{dt}\right)_{SBA,i,day} \cdot 365 \text{ days} \cdot A_i \cdot V_{atmos} \tag{5}$$

where $V_{atmos}$ is the volume of atmosphere ($3 \cdot 10^{18}$ m$^3$) and $A_i$ is the surface fraction of each ecosystem (***Table 2***). The production fluxes for each ecosystem are shown in ***Figure 3b***, together with their sum for all ecosystems. The total predicted amount of SBA production is 3.7 Tg yr$^{-1}$ with highest contributions from bacterial activities (0.5 Tg yr$^{-1}$ each) above seas as they cover most of the globe (71%), above shrubs (0.5 Tg yr$^{-1}$) since the highest bacteria concentrations have been identified there (***Table 1***) and above forests with a much smaller surface area (7%) but higher cell concentration. Using average data instead of those for individual ecosystems results in 0.7 Tg yr$^{-1}$. Given the large uncertainties in all factors of Eq (4) and Eq (5), we suggest that the value based on the weighted sum of all ecosystems (3.7 Tg yr$^{-1}$, Category 'All') might be a reasonable 'first best estimate' of total SBA contribution by bacteria on a global scale. This value are similar to the range of 1 - 10 Tg yr$^{-1}$ that was extrapolated by Sattler et al. (2001) based on carbon production rates of bacteria in supercooled clouds at mount Sonnblick observatory in the Austrian Alps.

Our estimated SBA mass production represents the production of total bacteria mass. The carbon content of bacteria cells is roughly 50% of their dry mass, with the remainder composed of nitrogen, oxygen, phosphorous, hydrogen and other elements (Whitman et al., 1998). Thus, we suggest that SBA formation may lead to ~1.9 Tg carbon yr$^{-1}$ based on our best estimate that is bound in biological mass.

**3.1.2 Discussion of uncertainties in SBA formation**

The formation rates in ***Section 3.1.1*** represent an estimate of a previously unrecognized source of biological aerosol mass in the atmosphere. All parameters are associated with large uncertainties that need to be constrained in the future as they might vary depending on temporal, meteorological, spatial and geographical conditions. Ranges of observed values for all parameters in Eq (4) and Eq (5) are summarized in ***Table 3*** and discussed in the following:

(i) The cell concentrations $C_{cell}$ in the atmosphere homogenize aloft due to mixing processes and average to concentrations of ~$10^4$ m$^{-3}$ at most locations. However, spatial deviations might be present in particular locations, such as cell concentrations of ~$7 \cdot 10^5$ - $4 \cdot 10^6$ cells m$^{-3}$ that were found during haze periods in China (Li et al., 2018; Xie et al., 2018), and even $10^9$ m$^{-3}$ above a wastewater storage lagoon which can be considered the highest expectable value of ambient bacteria (Paez-Rubio et al., 2005). Using the framework presented in the present study, SBA formation in such spatially limited areas can be estimated if growth rates of the individual bacteria types were available.

(ii) The growth rates $R_{cell}$ assumed here likely represent an overestimate as cloud temperatures are often lower than ~15-20°C. At temperatures > ~0°C, this overestimate is likely less than an order of magnitude (***Figure 2***); in supercooled cloud droplets (< 0°C), metabolic activity $R_{cell}$ might be some orders of magnitude lower and cell multiplication can be considered negligible.

(iii) While some of the studies listed by Burrows et al. (2009b) and in ***Table 1*** report the concentrations of viable cells, others give the total cell concentrations. In addition, the large discrepancy in reported $F_{live}$ between < 0.1% up to nearly 100% as discussed e.g., by Lindeman et al. (1982), Lighthart and Shaffer (1994) and Gandolfi et al. (2013), might be

also due to differences in the measurement techniques. Consistent experimental methodologies are needed to give comprehensive data on $F_{live}$ and the survival rates of bacteria in aerosol particles (Otero Fernandez et al., 2019).

(iv) The average cell mass depends on bacteria type and their growth stage. Sattler et al. (2001) estimated carbon mass of bacteria cells in cloud water as 17 fg carbon cell$^{-1}$ in agreement with values in marine and freshwater ecosystems. Approximating the total mass by doubling the carbon mass, results in 34 fg cell$^{-1}$, i.e. equivalent to spherical particles with diameters of ~0.4 μm. Carbon mass and the carbon-to-total mass ratio can greatly differ from these values; for example, total masses of prokaryotic cell of 200 fg cell$^{-1}$ in soil have been reported (Whitman et al., 1998). In their global study, Burrows et al. (2009b) assumed a mass of 520 fg cell$^{-1}$ (1 μm particle).

(v) While several studies have shown that liquid water is necessary for efficient microbial activity, it is not clear yet whether bacteria maintain activity in wet aerosols. Klein et al. (2016) found indications that bacteria metabolic activity exists in aerosols but no quantitative data is reported yet. Bacteria become dormant (Kaprelyants and Kell, 1993) or have reduced viability at relative humidities of 86 - 97% (Haddrell and Thomas, 2017). In soil samples, it has been shown that cycles of drying and rewetting might enhance microbial activity compared to constantly moist samples (Meisner et al., 2017; Xiang et al., 2008); thus, it may be speculated that such effects also occur in rapidly changing humidity conditions in atmospheric deliquesced aerosols. Under those conditions, the time fraction of microbial activity would exceed $F_{cloud}$. SBA mass formation calculated by Eq (4) and Eq (5) depends linearly on all parameters discussed in (i) to (v). Thus, the uncertainty of the predicted formation rates can be simply estimated by the ranges given in *Table 3*. However, in the atmosphere, all parameters might continuously change over time and thus might affect SBA mass to different extents.

### 3.1.3 Comparison of SBA formation to other aerosol sources

An estimate of aerosol emissions from the biosphere suggested a source strength of PBA mass of 1000 Tg yr$^{-1}$ (Jaenicke, 2005). However, in this latter study, PBA was defined to include all cellular material, proteins, and their fragments. A global model study predicted total PBA emissions (bacteria, fungal spores and pollen) of 123 Tg yr$^{-1}$, of which bacteria comprised 0.79 Tg yr$^{-1}$, fungal spores 5.8 Tg yr$^{-1}$ and pollen 47 Tg yr$^{-1}$ (Myriokefalitakis et al., 2017). These numbers are similar to the range of 0.4 – 1.8 Tg bacteria yr$^{-1}$ and pollen 47 Tg yr$^{-1}$ (Burrows et al., 2009b), and 31 Tg fungal spores yr$^{-1}$ (Hoose et al., 2010). However, as pollen grains have usually sizes > 30 μm (Winiwarter et al., 2009), their atmospheric residence time is limited and thus their burden to total PBAP is relatively smaller as compared to that of fungal spores (6.2 Gg vs 773.4 Gg, respectively) (Myriokefalitakis et al., 2017). It was suggested that the global emissions of fungal spores (25 Tg yr$^{-1}$) comprise 23% of total primary organic aerosol (Heald and Spracklen, 2009). A study based on tracer compounds resulted in an emission estimate for fungal spores of 50 Tg yr$^{-1}$ (Elbert et al., 2007). None of these estimates include microbial activity as a source of biological mass. Our predicted SBA source of 3.7 Tg yr$^{-1}$ is restricted to the mass production by bacteria but is similar to predictions for primary bacteria emissions. The estimates of primary bacteria emissions were performed such that observed cell concentrations are matched by models without considering another source of cells in the atmosphere (Burrows et al. (2009a,b), Myriokefalitakis et al. (2017)). Our SBA estimates might be equally biased as they are based on the same ambient cell concentrations that

might comprise emitted and new bacteria cells. The absolute values and the ratio of primary to secondary bacteria mass need to be evaluated by more complex model studies as our simple framework can provide.

Total organic aerosol is composed of mostly secondary mass. Best estimates based on observational and model studies of the net production rate of secondary organic aerosol (SOA) mass are on the order of $136 - 280$ Tg yr$^{-1}$ (Hodzic et al., 2016). These amounts are similar to the predicted global sulfate production of 117 Tg yr$^{-1}$ (39 Tg S yr$^{-1}$) (Chin et al., 2000). Thus, SBA production can be estimated to be on the order of ~1% of the secondary aerosol sources. The net aerosol mass formation due to SBA production might be even smaller if bacteria metabolize substrates that are already in the particle phase. In this case, biotransformation processes lead to the conversion of non-biological into biological aerosol mass. The unique properties of biological aerosol material have been extensively discussed in the context of heterogeneous ice nucleation where it has been shown that even small amounts of biological material could have significant effects on clouds and precipitation (Möhler et al., 2008; Morris et al., 2004; Šantl-Temkiv et al., 2015). Given the low ambient concentrations of ice nucleating particles and their high sensitivity to the ice/liquid partitioning in mixed-phase clouds (e.g., Ervens et al., 2011), a small change in biological mass possibly translates into significant changes in the evolution of cold clouds.

## 3.2 Consumption of organic carbon in clouds

### 3.2.1 Calculation of microbial and chemical WSOC loss rates

Bacteria can be metabolically active in the aqueous phase of clouds (Delort et al., 2010) and on the surface or bulk phase of particles (Klein et al., 2016; Estillore et al., 2016). These metabolic processes are typically enzyme-mediated chemical reactions within the bacteria cells that supply the necessary energy for the cells to maintain their vitality. The cells typical utilize small organic compounds leading to a decrease of water-soluble organic carbon (WSOC) mass within cloud droplets as bacteria convert these substrates into $CO_2$ (*Figure 1*). Also processes of biological mass production from $CO_2$ exist (autotrophy) and include photosynthesis (photoautotrophs). However, despite the fact that photosynthetic microorganisms were reported in the atmosphere (Tesson and Šantl-Temkiv, 2018) there is neither a clear indication yet of photosynthetic activity in clouds, nor of other modes of autotrophy.

The split between carbon uptake for biological mass production and mineralization is quantified by the bacterial growth efficiency BGE (Eq (2)). Studies have shown that generally metabolic processes produce mostly $CO_2$ under atmospheric conditions and only a small fraction of the carbon is mineralized into organic products ($< 1 - 10\%$ of the total C utilized). We introduce here the factor $F_{CO2}$ as a measure of the loss of organics due to bacterial processes:

$$F_{CO2} = 1 - BGE \tag{6}$$

Using $F_{CO2}$, we can write the carbon balance as

$$WSOC \xrightarrow{Bacteria} F_{CO2}\ CO_2 + (1 - F_{CO2})\ WSOC \tag{R1}$$

The loss rate of carbon can be calculated as

$$R_{WSOC,Bact} \left[ \frac{g}{L(aq)\ s} \right] = -\frac{d(WSOC)}{dt} = \frac{dCO_2}{dt} =$$

(7)

$$-F_{CO2}\ k_{Bact} \left[ \frac{L(aq)}{cell \cdot s} \right] \cdot C_{Cell,aq} \left[ \frac{cell}{L(aq)} \right] \cdot F_C \cdot C_{WSOC} \left[ \frac{g_C}{L(aq)} \right] \cdot F_{cloud}$$

whereas the cell concentration in cloud water can be replaced by

$$C_{Cell,aq} \left[ \frac{cell}{L(aq)} \right] = C_{cell,g} \left[ \frac{cell}{m^3(g)} \right] / LWC \left[ \frac{m^3(g)}{L(aq)} \right]$$

(8)

with $C_{Cell,aq}$ and $C_{WSOC}$ being the concentrations of bacteria and water-soluble organic carbon in cloud water, respectively, and $C_{cell,g}$ ambient cell concentrations in the gas phase (e.g. **Table 1**). These concentrations are on average $C_{Cell,aq} \sim 10^7$ cells L$^{-1}$ (Vaïtilingom et al., 2013) and $C_{WSOC} \sim 0.1$ mM (Herckes et al., 2013) in cloud water whereas both values might differ over a few orders of magnitude locally and temporally (**Section 3.2.2**). Usually experimental loss rates of organics by bacteria in real and artificial cloud water are reported to be on the order of $\leq 10^{-17}$ mol cell$^{-1}$ s$^{-1}$ for organic substrates (e.g., formic, acetic, and succinic acids) (Vaïtilingom et al., 2010, 2011). The cell activity is dependent on the bacteria type and the availability of the organic substrate. Thus, strictly, such rates [mol cell$^{-1}$ s$^{-1}$] are only valid for the substrate-to-cell ratio as applied in the experiments. In order to account for the ratio as encountered in cloud water, we use here $k_{Bact}$ [L cell$^{-1}$ s$^{-1}$] (i.e. measured rate divided by the concentration of organic substrate in the experiments) that is applicable to the full range of conditions where the cells exhibit a similar microbial activity. Resulting rate constants for formic, acetic and succinate acids are on the order of $k_{Bact} \sim 10^{-13}$ L cell$^{-1}$ s$^{-1}$. Equation 7 includes an additional factor $F_C$ that accounts for the microbial selectivity towards only some organics by each bacteria type (e.g. Šantl-Temkiv et al., 2013; Bianco et al., 2019). For example, it has been shown in a single study that, upon laboratory incubation of cloud water, oxalic acid is not affected by cloud borne microorganisms, formate is only consumed after a lag time of several hours, which is much longer than the lifetime of a cloud droplet, and compounds such as acetate or succinate are readily biodegraded (Vaïtilingom et al., 2011). Since these compounds comprise major constituents of WSOC in cloud water, it seems reasonable to introduce a factor $F_C < 1$. Given the complexity of the organic matter in the atmosphere, the numerous organic molecules existing in cloud water and their variable susceptibility to biodegradation, $F_C$ is hard to specify with precision. Bianco et al. (2019) observed experimentally by FT-ICR-MS during laboratory incubation of cloud water that of the 2178 compounds detected, 1094 were utilized by bacteria (~50%). Assuming that all these compounds were equally abundant, one could conclude that 50% of all cloud water organics were prone to be microbiologically consumed (i.e. $F_C = 0.5$). More quantitative support of this assumption could be given based on the fact that preferably small oxygenated organics are taken up by bacteria. Compilations **o**f speciated cloud water organics have shown that small acids, such as formic and acetic acid, comprise a large fraction (at least ~30%) of the characterized fraction of cloud water organics (e.g. Figure 6 in Herckes et al., (2013). The calculated rates $R_{Bact,WSOC}$ are summarized in **Figure 4** for $0.8 \leq F_{CO2} \leq 0.99$ and $F_C = 0.5$. While $F_C = 0.5$ seems a reasonable compromise, this factor is highly uncertain and strongly depends on the microbial and chemical composition of cloud water.

Several studies have discussed the competition of microbial and chemical processes in cloud water as a sink of specific organic compounds (Ariya et al., 2002; Husárová et al., 2011; Vaïtilingom et al., 2010, 2013). The most efficient loss reactions for organics in cloud water are initiated by OH radicals. The general rate constant of the OH radical with water-soluble organic carbon is $k_{OH} = 3.8 \cdot 10^8$ M$^{-1}$ s$^{-1}$ (Arakaki et al., 2013). The reactions of WSOC with OH lead to

volatile and non-volatile oxidation products. Radicals are much less selective towards organics than bacteria are; thus, the assumption of a factor equivalent to $F_C$ in Eq (7) is not necessary as all water-soluble organics react with OH with the chemical reactivity mostly depending on the structure of the organic compound. The yield of volatile products ($Y_{volC}$) includes $CO_2$ but also formaldehyde and other volatile compounds that do not remain in the particle phase after cloud evaporation and thus do not contribute to the aerosol loading. We assume $0.2 \leq Y_{volC} \leq 0.5$, but in general, $Y_{volC}$

depends on the WSOC composition, with higher values for more aged organics that are more readily oxidized to volatile products. This upper limit might be representative, for example, for fog water as characterized in Fresno (CA, USA) where about 50% of the dissolved organic carbon was composed of small acids (formic, acetic, oxalic) and aldehydes (formaldehyde, dicarbonyls), small aldehydes are oxidized in the aqueous phase to the carboxylic acid; oxidation of carboxylic acids yields $CO_2$ (Ervens et al., 2003).

Equivalent to reaction (*R 1*), we express the carbon loss by the OH radical in clouds as

$$\text{WSOC} + \text{OH} \rightarrow Y_{volC} \text{ Volatile Products} + (1 - Y_{volC}) \text{ WSOC}_{aer} \tag{R2}$$

With WSOC$_{aer}$ the WSOC fraction that remains in the aerosol phase after drop evaporation. We calculate the loss rate accordingly:

$$R_{OH,WSOC} = -\frac{d(\text{WSOC})_{OH}}{dt} \left[\frac{g}{L(aq)s}\right]$$

$$= k_{OH} [L\ mol^{-1}\ s^{-1}]\ [\text{OH}]_{aq} [mol\ L^{-1}] \cdot Y_{volC} C_{WSOC} \left[\frac{g_C}{L}\right] \cdot LWC \left[\frac{g}{m^3}\right] \cdot F_{cloud} \tag{9}$$

OH concentrations in cloud water are in the range of $10^{-16}$ M $< [\text{OH}]_{aq} < 10^{-14}$ M (Arakaki et al., 2013; Bianco et al., 2015; Ervens et al., 2014) and an average cloud liquid water content (LWC) of 0.15 g m$^{-3}$ is assumed. The results in *Figure 4* suggest that the microbial rates may be comparable to or smaller than the chemical ones under most conditions.

Overall, the values shown in Figure 4 only differ by a factor of ~2.5 which might imply that there are conditions under which chemical and biological processes in the aqueous phase compete. This trend is in agreement with several previous studies that focused on the comparison of microbial versus chemical processes as sinks for specific organic substrates (Amato et al., 2007a; Vaïtilingom et al., 2010). These loss fluxes are relatively large as compared to the predicted SBA formation (*Figure 3*).

In a previous study, it was estimated that microbial processes in clouds lead to a total carbon loss of ~10 – 50 Tg yr$^{-1}$ and to a production of ~100 Tg yr$^{-1}$ $CO_2$ with the assumptions of complete respiration ($F_{CO2} = 1$), microbial non-selectivity towards WSOC ($F_C = 1$) and applying the same loss rates as observed in lab experiments without correcting

for differences in the ratio of bacteria cell to WSOC concentrations (Vaïtilingom et al., 2013). Thus, this former estimate can be considered an upper limit whereas the one in the current study (~30 Tg yr$^{-1}$) is more conservative, suggesting that the respiration of bacteria is a negligible global $CO_2$ source as compared to the sum of anthropogenic sources (~50,000 Tg $CO_2$ yr$^{-1}$ (IPCC, 2014)). We can conclude that the loss of WSOC by chemical and biological processes is relatively small (~10%) compared to the total removal of water-soluble organic carbon from the atmosphere by wet deposition derived from global models (293 Tg yr$^{-1}$ (Safieddine and Heald, 2017); 306 Tg yr$^{-1}$ (Iavorivska et al., 2016; Kanakidou et al., 2012)).

**3.2.2 Discussion of uncertainties of microbial and chemical WSOC loss**

The calculation of microbial and chemical WSOC loss should be regarded an approximation using a set of parameters that are all associated with considerable uncertainties and variability depending on the bacteria and cloud characteristics. It can be expected that on small spatial and temporal scales, all parameters vary significantly. Similar to the discussion in *Section 3.1.2*, we compile all parameters and atmospherically relevant minimum and maximum values based on literature data in *Table 4*. Equation (7) is a linear equation; thus, a change in any of the parameters will translate into a proportional change in predicted WSOC loss

(i) There are not as many measurements of $C_{Bact,aq}$ as for cell concentrations $C_{cell}$ in cloud-free regions. The assumption that all particles that contain bacteria cells are activated into cloud droplets does likely not lead to a large overestimate. The sizes of bacteria-containing particles usually exceed several hundred nanometers and thus can all be considered CCN. Differences in LWC - as the conversion factor from gas to aqueous phase concentrations - are relatively small, within a factor of 2 – 3, within the categories of common cloud types (Pruppacher and Klett, 2003).

(ii) The activity of microorganisms towards organic substrates is often reported in units of 'mol(substrate) cell$^{-1}$ s$^{-1}$' which expresses the amount of substrate that is consumed per cell and time. For several compounds (e.g., formate, acetate, succinate) these rates differ by approximately one order of magnitude (Vaitilingom et al., 2011). However, the resulting $k_{Bact}$ values (i.e. rate divided by substrate concentration) are all on the order of $k_{Bact} \leq 10^{-13}$ L cell$^{-1}$ s$^{-1}$ which appears to represent an upper limit for the organics that have been investigated for metabolic activity in clouds. A much lower constant was derived from experiments with less oxygenated compounds such as phenol (Lallement, 2017).

While we only consider the direct interaction of bacteria and organics, additional processes might lead to more complex chemical and microbial interactions. For example, siderophores form iron complexes (Passananti et al., 2016) and, thus, suppress Fenton reactions that affect oxidant levels in cloud droplets (e.g., Deguillaume et al., 2004). Such indirect feedbacks of microbial processes on chemical budgets require more comprehensive data sets that are currently not available for models.

(iii) The respiration of bacteria depends on many different factors such as stress due harsh conditions. It can be expected that at higher stress levels (nutritional or thermal), $F_{CO2}$ increases to supply elevated energy needs (Amato and Christner, 2009; Eiler et al., 2003).Values of BGE as low as < 0.4% ($F_{CO2}$ = 0.996) were observed (Eiler et al, 2003 and references therein), indicating that nearly all carbon used was mineralized into $CO_2$.

(iv) The fraction of organic material metabolized by bacteria in clouds ($F_C$) is likely not unity for a single bacteria type (Bianco et al., 2019; Vaïtilingom et al., 2011). Carboxylic acids that are preferentially metabolized by several common bacteria types often comprise a major fraction (~ 20%) of the cloud-water organics that can be speciated on a molecular level (Herckes et al., 2013). This fraction might be regarded a lower limit of $F_C$ since the reactivity of the large fraction of unspeciated organics (often ~70%) towards bacteria is not known. However, a recent qualitative study suggested that ~50% of all organics in cloud water are microbially consumed by bacteria (Bianco et al., 2019). Our comparison implies the same spatial accessibility of bacteria and OH, respectively, to WSOC. This might be an oversimplification as bacteria are unevenly distributed among cloud drop populations as statistically only one in ~10,000 droplets may contain a single bacteria cell. OH can be expected to be present in all cloud droplets as the direct phase transfer from the gas phase represents one of the major OH sources in cloud water.

(v) While the absolute importance of microbial loss depends on the parameters discussed in (i) to (iv), the relative importance compared to chemical processes might be of interest in studies where the fate of individual organics in the cloud droplets or in the atmospheric multiphase system is explored. $[OH]_{aq}$ depends mostly on photochemical processes as source processes and on the concentrations of WSOC as the main sinks; it ranges from $10^{-17}$ M (night-time) to $10^{-14}$ M (day time, clean air masses) (Arakaki et al., 2013).

(vi) Given that formate and acetate comprise major contributors to cloud water organics (Herckes et al., 2013), some fraction of WSOC will be converted into highly volatile products, such as $CO_2$ and $CH_3CHO$ that will not remain in the particle phase after cloud evaporation. However, $Y_{volC}$ likely does not approach unity since several studies have suggested that radical reactions in cloud water lead to the successive decay of dicarboxylic acids into their next smaller homologue which will remain in the aqueous phase. Within these limits, we conservatively suggest a range of 0.2 to 0.8 for $Y_{volC}$ but point out the need for studies to refine this parameter

## 4.  Summary and conclusions

We have estimated the amount of biological mass that is formed in the atmosphere by growth and multiplication of bacteria cells ('secondary biological aerosol', SBA). Data for representative bacteria strains and their generation rates have been compiled for major ecosystems. Using average values for cloudiness above the various ecosystems, we estimate that 3.7 Tg yr$^{-1}$ SBA mass is formed globally which is comparable to current estimates of direct bacteria emissions (0.4 – 0.7 Tg yr$^{-1}$ (Burrows et al., 2009b; Myriokefalitakis et al., 2017)) which comprise a small fraction of total biological aerosol mass. While these production rates make up ~1% of other major secondary aerosol formation rates (secondary organics or sulfate), their importance might differ on spatial or temporal scales. In addition, SBA production leads to an increase in biological aerosol mass which might sensitively affect physicochemical particle properties (e.g. ice nucleation ability). SBA formation linearly depends on several parameters, such as the number concentration of metabolically active bacteria cells, their generation rates and the time scales during which they are assumed to grow or multiply – all of which are associated with considerable uncertainties. The ecosystem categories in Table 2 represent fairly large regions. It might be expected that SBA formation rates are different on smaller spatial and/or temporal scales. For example, it has been shown that human activities in cities lead to high bacteria

concentrations; also forests have been identified as significant sources of biogenic aerosol. However, detailed data on bacteria are sparse in such regions. While several recent studies have characterized the diversity of microorganisms in forested regions (rainforest, tropics) (Gusareva et al., 2019; Souza et al., 2019), these studies did not report cell concentrations which highlights the urgent need of additional measurements.

The detailed discussion of the parameters and their uncertainties in our simplified approach highlights the likely variability of SBA formation on smaller scales and the need of future studies to refine these parameters. Similar approaches as ours may be applied to yeast growth. Yeast cells are generally larger ($\sim 2 - 10$ µm) than bacteria cells (Fröhlich-Nowoisky et al., 2009) and, thus, their residence time in the atmosphere is likely shorter. Detailed data on their activity in clouds are not available which currently prevents the assessment of their potential contribution to SBA.

We also quantify the role of clouds as sinks of total WSOC by microbial and chemical processes, unlike previous studies that focused on microbial activity towards individual organic compounds. It is estimated that microbial processes lead to an organic mass loss of $8 - 11$ Tg yr$^{-1}$ whereas chemical processes by the OH radical in clouds lead to a loss of $8 - 20$ Tg yr$^{-1}$. These numbers are small compared to other sinks such as aerosol removal by deposition. Not all of the WSOC mass even contributes to organic aerosol loading as water-soluble, volatile organics are dissolved

in cloud water but evaporate during drop evaporation. Thus, the loss of organic aerosol mass due to direct microbial activity in clouds might be smaller than the predicted loss of WSOC. Large uncertainties in these estimates represent the assumptions on the fraction of carbon that is converted into volatile products. For bacteria, this fraction is quantified by the bacteria growth efficiency that depends on numerous factors, such as bacteria type, substrate availability and physical conditions in the condensed phase.

In current atmospheric models, when considered, bacteria cells are inert, i.e. they neither change their mass or number concentrations during their residence time in the atmosphere nor do they interact with other aerosol constituents. The approach presented in our study provides a first simplified estimate of SBA formation and WSOC loss due to bacteria that could be easily adapted in models. Given the current great activities in the field of atmospheric bioaerosols, it can be expected that the discussed parameters in the estimates can and should be refined in the future in order to quantify

the role of bacterial processes as source of biological mass and net source or sink of organic aerosol in the atmosphere.

**Data availability.** This study is based on literature data; all sources are reported in the text. All results that are discussed were obtained by Equations (4) – (9), using the data in Table 1, are displayed in Figures 3 and 4. No additional data can be reported.

**Author contributions.** BE and PA planned and carried out the study and wrote the manuscript together.

**Competing interests.** The authors declare that they have no conflict of interest.

**Acknowledgement.** This work has been funded by the French National Research Agency (ANR) in the framework of the 'Investment for the Future' program, ANR-17-MPGA-0013.

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

**Table 1:** Summary of ambient cell concentrations $C_{cell}$ and generation rates $R_{cell}$ for predominant bacteria types in all ecosystems

| | $C_{Cell}$ [m$^{-3}$] [a] | Representative strain affiliation | Generation rate $R_{Cell}$ [h$^{-1}$] |
|---|---|---|---|
| All | 10,000 | Alpha-Proteobacterium *Sphingomonas* sp. (average of 32b-11, 32b-49, 32b-57, 35b-32, 35b-38) [e] | 0.06 (5°C)[e] |
| | | | 0.2 (17°C)[e] |
| | | | 0.35 (27°C)[e] |
| | | | 0.45 (37°C)[e] |
| Tundra | 12,000 | *Pseudomonas* spp. (*P. graminis*) [e, f, g] | 0.12 (5°C)[e] |
| | | | 0.21 (17°C)[e] |
| | | | 0.82 (27°C)[e] |
| | | | 0.27 (37°C)[e] |
| | | *Psychrobacter* sp.[g] | 0.0007 (-10°C)[g] |
| | | *Rhodococcus* sp.[g] | 0.0001 (-10°C)[g] |
| Grassland | 110,000 | *Pseudomonas syringae* [e, h] | 0.1 (5°C)[e] |
| | | | 0.25 (15°C)[e] |
| | | | 0.9 (27°C)[e] |
| Coastal | 76,000 | | |
| Wetlands | 90,000 | | |
| Crops | 110,000 | *Frigoribacterium* sp. [e, i] | |
| Land ice | (5,000) | *Raphidonema* spp.[j] | 1.7·10$^{-4}$ -2.91·10$^{-4}$ (12-18°C)[j] |
| Deserts | (10,000) | | |
| | 612 [c] | | |
| Forests | 56,000 | | |
| | 6,323-12,748 [d] | | |
| Shrubs | 350,000 | | |
| Seas | 10,000 | *Pseudoalteromonas* [k] | 0.25 (T unknown) [k] |
| Seas (estuary) [b] | | Gamma-Proteobacterium (the fastest) [l] | 0.17 (14°C) [l] |
| | | | 0.19 (24°C) [l] |

[a] All cell concentrations are taken from Burrows et al. (2009b) unless otherwise noted. [b] 'seas estuary' was not included as a separate ecosystem by Burrows et al. (2009b); [c] (Lighthart and Shaffer, 1994); [d] (Helin et al., 2017); [e] Amato et al. (unpublished data; strains originally reported in Amato et al., 2007c and Vaïtilingom et al., 2012); [f] (Männistö and Häggblom, 2006); [g] (Bakermans et al., 2003); [h];(Morris et al., 2000) [i] (Copeland et al., 2015); [j] (Stibal and Elster, 2005); [k] (Middelboe, 2000) ; [l] (Fuchs et al., 2000)

**Table 2:** Surface coverage of ecosystems on Earth surface (Burrows et al., 2009b) ), approximate cloud coverage $F_{clc}$ above the ecosystems, estimated based on maps of cloud cover data obtained by MODIS Terra for spring (2000-2011), and estimated time fraction bacteria spend in clouds ($F_{cloud}$)

|           | % of Earth surface [a] | $F_{clc}$ | $F_{cloud}$ |
|-----------|:----------------------:|:---------:|:-----------:|
| All       | 100                    | 0.6       | 0.15        |
| Tundra    | 3.3                    | 0.4       | 0.1         |
| Grassland | 2.2                    | 0.7       | 0.2         |
| Coastal   | 0.2                    | 0.4       | 0.1         |
| Wetlands  | 0.6                    | 0.5       | 0.15        |
| Crops     | 3.0                    | 0.7       | 0.3         |
| Land ice  | 3.1                    | 0.4       | 0.1         |
| Deserts   | 3.7                    | 0.2       | 0.05        |
| Forests   | 7.0                    | 0.9       | 0.25        |
| Shrubs    | 5.8                    | 0.3       | 0.1         |
| Seas      | 71.0                   | 0.7       | 0.2         |

[a] Data from Burrows et al. (2009b) and Olson *(1992)*

**Table 3:** Parameters used in the estimate of SBA mass formation and their possible minimum and maximum values based on literature data

| Parameter | Value in Eq (4) and Eq (5) | Range | | Comment |
|---|---|---|---|---|
| | | Minimum | Maximum | |
| $C_{cell}$ [m$^{-3}$] | $10^4$ | 100 | $10^9$ | $C_{cell,min}$: above desert during low RH and high radiation (Lighthart and Shaffer, 1994); ~$10^6$: in a highly polluted area (Xi'an, China) (Xie et al., 2018); $C_{cell,max}$ was measured above a wastewater storage lagoon (Paez-Rubio et al., 2005) |
| $R_{cell}$ [h$^{-1}$] | 0.3 | 0 | 3 | $R_{cell} \sim 0$ might occur under stressful conditions when cells become dormant (e.g. low temperature, little water); $R_{cell} \sim 3$ for *E. coli* under optimal conditions (37°C, appropriate culture medium). |
| $F_{live}$ | 1 | 0.0001 | 1 | 0.0001 – 0.2 based on global microbial diversity (Gandolfi et al., 2013); 0.22 above crop fields (Lindemann et al., 1982), 0.81 above desert (Lighthart and Shaffer, 1994). Some studies report concentrations of viable cells; in this case, these concentrations imply $F_{live} = 1$ |
| $m_{cell}$ [fg cell$^{-1}$] | 52 | 34 | 520 | $m_{cell,min}$ corresponds to cells in clouds assuming that cells are composed of 50% carbon (Sattler et al., 2001); masses of other prokaryotic cells might be ≤200 fg cell$^{-1}$ (Whitman et al., 1998); $m_{cell,max}$ corresponding to a spherical cell of a diameter 1µm and density of 1 g cm$^{-3}$ (Burrows et al., 2009b). |
| $F_{cloud}$ | 0.15 | > 0 | 1 | The average global value might be higher than 0.15 if bacterial processes also occur outside of clouds. On small scales or above individual ecosystems, the value for clouds can be smaller or larger than the average value, depending on cloud variability. |

**Table 4:** Values for parameters in Eq (7) and Eq (9) used in the estimate of WSOC loss by microbial and chemical processes and their minimum and maximum values

| Parameter | Value in Eq (7) and Eq (9) | Range | | Comment |
| | | Minimum | Maximum | |
|---|---|---|---|---|
| $C_{Bact,aq}$ [cells $L^{-1}$] | $10^7$ | $10^6$ | $10^8$ | Range of total bacteria concentration observed in 31 cloud water samples collected from a mid-altitude mountain site over several years (Vaïtilingom et al., 2012) |
| LWC [g $m^{-3}$] | 0.15 | 0.1 | 1 | The minimum and maximum value describe a range for a wide variety of cloud types. The assumption of LWC is not needed if it is assumed that all bacteria-containing particles act as CCN. |
| $k_{Bact}$ [L $cell^{-1}$ $s^{-1}$] | $10^{-13}$ | $10^{-15}$ | $10^{-13}$ | $k_{Bact,min}$ was derived for microbial activity towards phenol (Lallement, 2017). $k_{Bact,max}$ was derived from experiments using cloud water and is valid for the microbial activity of various highly oxygenated compounds. |
| $F_{CO2}$ | 0.8 – 0.99 | 0.2 | 1 | Even though BGE ranging from < 0.4% to 80% (0.996 > $F_{CO2}$ > 0.2) were estimated in natural environments (Eiler et al. 2003 and references therein), at low nutrient concentrations, as encountered in clouds, high $F_{CO2}$ can be expected. |
| $F_C$ | 0.5 | 0.2 | < 1 | Herckes et al. (2013) report that ~20% of total organic carbon in clouds is composed of speciated carboxylic acids; Bianco et al. (2019) demonstrate that ~50% of all organics in cloud water are affected by bacteria. $F_C = 1$ seems unlikely due to variation in microbial and chemical cloud water composition. |
| $k_{OH}$ [L $mol^{-1}$ $s^{-1}$] | $3.8 \cdot 10^8$ | $10^6$ | $10^{10}$ | Typically, undissociated acids (low pH) and polyfunctional compounds have $k_{OH}$ at the lower end of this range whereas the upper limit is constrained by diffusion limitation (Herrmann, 2003; Monod and Doussin, 2008) |
| $[OH]_{aq}$ [M] | $10^{-15}$ | $10^{-17}$ | $10^{-14}$ | The suggested range includes concentrations that were inferred for night-time conditions (minimum) to day-time conditions in clean air masses (low OH sinks). |
| $Y_{volC}$ | 0.3 – 0.5 | 0.2 | 0.8 | This value has not been comprehensively quantified yet; largest values can be likely expected in aged WSOC with high O/C ratios |

**Figure captions**

**Figure 1:** Bacterial processes in the atmosphere leading to SBA formation and loss of water-soluble organic carbon (WSOC) in clouds.

**Figure 2:** Temperature dependence of generation rates $R_{cell}$ for bacteria types representative for the ecosystems in *Table 1* and *E coli* as a likely upper limit for cell generation. Dashed lines are extrapolations of the rates at T ~20°C using $Q_{10} = 2$ or $Q_{10} = 3$, respectively.

**Figure 3***:* Predicted production of secondary biological aerosol mass above the various ecosystems. The blue bar indicates the predicted production using the average values for all ecosystems ('all' in Table 1); a) SBA formation rates [ng m$^{-3}$ day$^{-1}$]; b) SBA production rates [Tg yr$^{-1}$]. The red shaded bar represents the sum of the contributions from all ecosystems, bacteria emissions are shown for comparison, taken from Burrows et al. (2009b)

**Figure 4:** Predicted loss of WSOC by bacterial utilization and by chemical (OH) processing in cloud water for different assumption on $F_{CO2}$ and $Y_{volC}$,

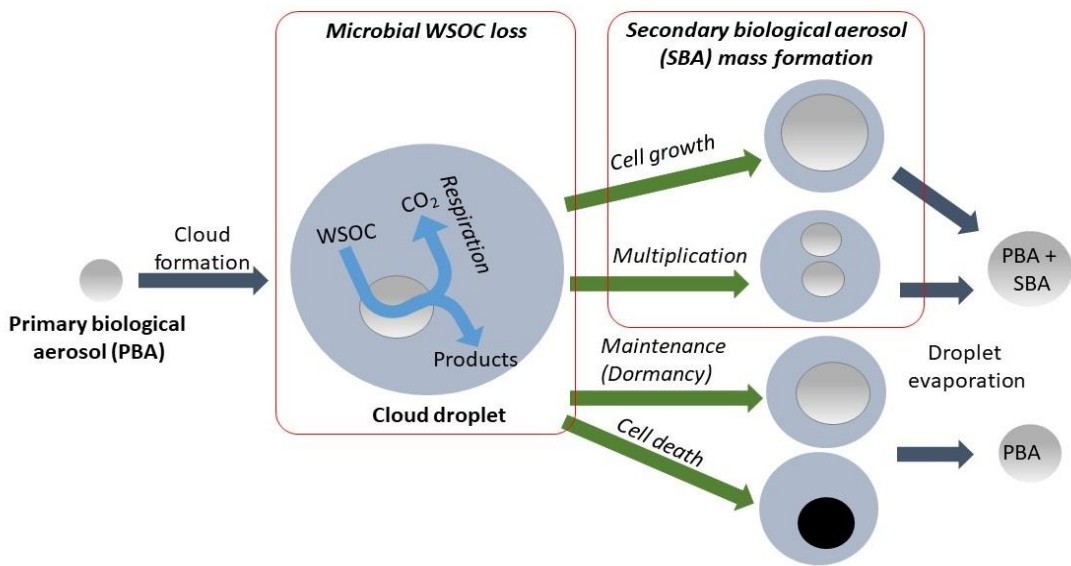

**Figure 1:** Bacterial processes in the atmosphere leading to SBA formation and loss of water-soluble organic carbon (WSOC) in clouds.

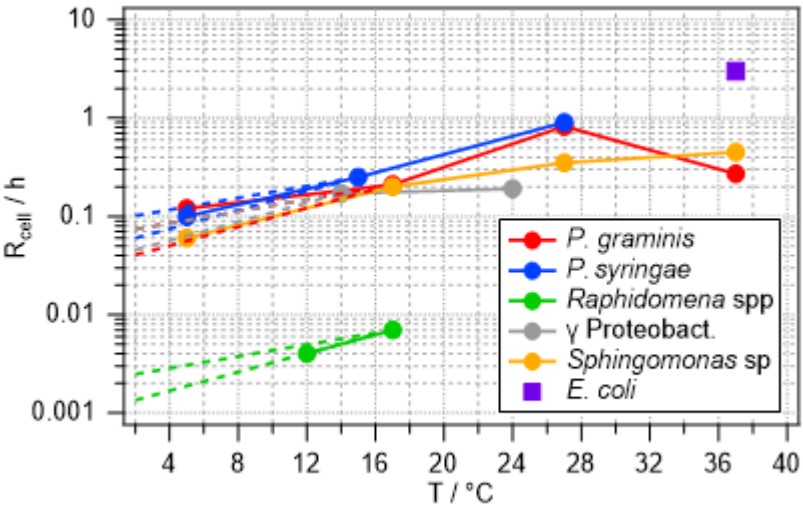

**Figure 2:** Temperature dependence of generation rates $R_{cell}$ for bacteria types representative for the ecosystems in *Table 1* and *E coli* as a likely upper limit for cell generation. Dashed lines are extrapolations of the rates at T ~20°C using $Q_{10} = 2$ or $Q_{10} = 3$, respectively.

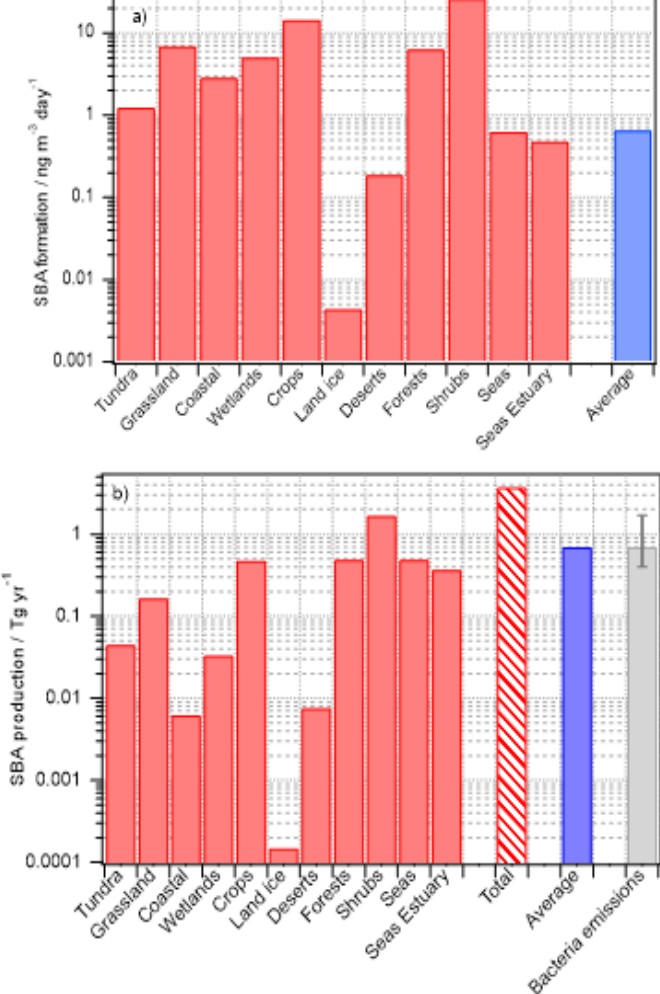

**Figure 3:** Predicted production of secondary biological aerosol mass above the various ecosystems. The blue bar indicates the predicted production using the average values for all ecosystems ('all' in Table 1); a) SBA formation rates [ng m⁻³ day⁻¹]; b) SBA production rates [Tg yr⁻¹]. The red shaded bar represents the sum of the contributions from all ecosystems, bacteria emissions are shown for comparison, taken from Burrows et al. (2009b)

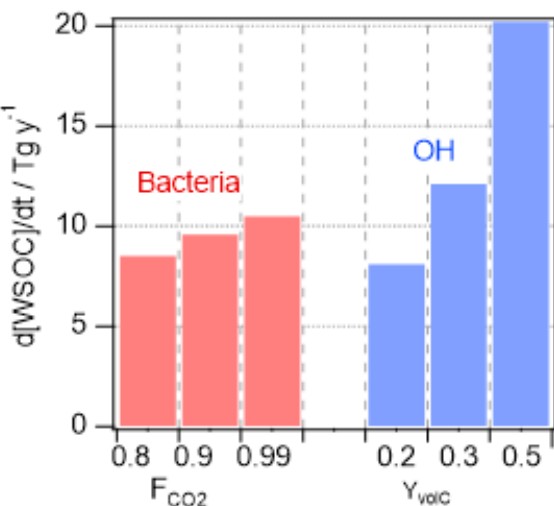

**Figure 4:** Predicted loss of WSOC by bacterial utilization and by chemical (OH) processing in cloud water for different assumption on $F_{CO2}$ and $Y_{volC}$,