# Peer review of "The global impact of bacterial processes on carbon mass"

_Atmospheric Chemistry and Physics, 2019_

## Referee Comment (RC1) · Anonymous Referee #3 · 14 Oct 2019

This is a very interesting and important study that identifies and tackles a major gap in the aerosol cloud interactions, that is already lacking primary biological particles to a large extent, and comes up with some rough estimates of the secondary biological particles. I find the paper suitable to be published in ACP, given that some issues raised below are answered.

1) How about sea-ice? Is it considered together with land-ice or not considered at all?

2) Similarly, urban sources? Why are not hey represented as they can be a large source of bacteria due to human existence?

3) Is it possible to provide with a formula that calculates Fcloud based on ecosystem, corresponding cloud fraction from MODIS and the conversion factor in order to be able

to reproduce the values in Table 2? Table 2 can be updated to include the cloud fraction over each ecosystem.

4) Table 1 caption in section 2.4.2 should be corrected to Table 2.

5) Is it possible to distinguish the different forest types or regions? It would be interesting to see these numbers above the amazons and boreal forests for example. Therefore, it would be interesting to show that global spatial distribution of this SBA source.

6) Line 236: 1% of the secondary aerosols.

7) Line 343: Where does the Fc=0.5 value come from, any reference or argument?

---

## Referee Comment (RC2) · Maria Kanakidou (Referee) · 17 Oct 2019

This is a very interesting paper that provides an innovative view – a new concept - of bacteria in the atmosphere. The authors make the point that a fraction of bacteria in the atmosphere is not inert but can multiply producing secondary bioaerosols and in parallel they can consume water soluble organic mass thus providing an alternative to the chemical degradation path for organic mass.

This is a nice and holistic view for the lifecycle of bacteria in the atmosphere. However, there are several gaps of knowledge in this cycle that the authors thoroughly discuss. Based on available literature and a number of assumptions that are clearly stated in the manuscript, the authors make rough calculations to evaluate the two terms involved

in the bacteria budget, namely the secondary production of bacteria and the bacteria driven degradation of water soluble organic mass. This later, the biological degradation of organic mass, is compared to the chemical degradation pathway and found of potentially similar importance but with a large range of uncertainty.

Despite the significant gaps in knowledge that prohibit accurate estimates, I consider that, overall, the manuscript provides a new concept for the presence and functioning of bacteria in the atmosphere that deserves publication in ACP after some improvements.

Abstract: lines 21-22: ' the conditions under which microbial processes cannot be neglected as organic carbon sinks in clouds' Please provide such information in the abstract.

Section 2.4.2: It is unclear which year of MODIS cloud data has been used. It would be nice to show the derived map of cloud volume and the ecosystem map that are later used to derive the numbers in Table 2 (in page 8 that is erroneously numbered as Table 1). What grid size is used for these calculations? How many grid points are used to derive the Fcloud over each ecosystem type? This is an important Table for the budget estimates that are further presented in the manuscript; therefore, it has to be well documented. Adding also a column with cloud fraction over each ecosystem as suggested by the other reviewer will be a significant improvement. It is also unclear 1) how the value of 0.15 is derived for FCloud, 2) whether the category 'Seas' in Table 2 contains also the sea-ice.

Line 215:explain to what you refer when writing 'other formation rates'

Caption of figure 3. The reference to Burrows et al is incomplete.

Line 273: 'sensitivity' I think 'uncertainty' is more appropriate here.

Table 3: maximum range for Ccell (1E9) can you comment who and where has measured this ?

Table 3: comment for mcell first line: 'assuming that they' to avoid confusion replace

[Figure]

'they' by 'cells'.

Lines 283-285: discussion about fungal spores: I do not see why this is discussed here. Please remove or rephrase sentence to better fit in the discussion.

Equation 6: Fc needs to be defined earlier, now it is defined in line 339. Also check units in this equation.

Line 340-342: provide an uncertainty range for Fc instead of one value. How this uncertainty is affecting the here presented estimates of WSOC loss by bacteria?

Line 364: 'slightly higher contribution of chemical reactions to WSOC loss'. Figure 4 shows that for Yvoc equal 0.5 the chemical loss can be double the bacteria loss. This is not 'slightly higher'. Please rephrase.

Line 372: Loss rate of 50 Tg/yr is stated here while in Figure 4, a maximum of about 30 Tg/yr is calculated. Make consistent.

Line 377: the authors claim that the WSOC losses are smaller than the predicted production rates of SOAaq. However, when accounting for the range of these rates, there is no significant difference. Furthermore, this result will depend on the assumed Fc, so please rephrase.

Line 433: I think 0.7 Tg/yr should be 3.7 Tg/yr

Line 458: in our study provides.

In addition, please provide references for 1) Figure 1 in its caption 2) the value of 7 to 14% in Line 159

---

## Author Comment (AC1) · 7 Nov 2019

The comment was uploaded in the form of a supplement:
https://www.atmos-chem-phys-discuss.net/acp-2019-619/acp-2019-619-AC1-supplement.pdf

---

## Author Response (AR1)

**Response to referee reports**

We thank Maria Kanakidou and the anonymous referee for their constructive comments on our manuscript and their careful referee reports. We respond to their comments below. Referee comments are in blue and our responses in black; modified text is shown in red. All line numbers refer to the revised manuscript version without annotations.

**Maria Kanakidou (Referee)**

This is a very interesting paper that provides an innovative view – a new concept – of bacteria in the atmosphere. The authors make the point that a fraction of bacteria in the atmosphere is not inert but can multiply producing secondary bioaerosols and in parallel they can consume water soluble organic mass thus providing an alternative to the chemical degradation path for organic mass. This is a nice and holistic view for the lifecycle of bacteria in the atmosphere. However, there are several gaps of knowledge in this cycle that the authors thoroughly discuss. Based on available literature and a number of assumptions that are clearly stated in the manuscript, the authors make rough calculations to evaluate the two terms involved in the bacteria budget, namely the secondary production of bacteria and the bacteria driven degradation of water soluble organic mass. This later, the biological degradation of organic mass, is compared to the chemical degradation pathway and found of potentially similar importance but with a large range of uncertainty. Despite the significant gaps in knowledge that prohibit accurate estimates, I consider that, overall, the manuscript provides a new concept for the presence and functioning of bacteria in the atmosphere that deserves publication in ACP after some improvements.

We thank the Maria Kanakidou for her positive comments on our manuscript. All comments are addressed in detail below.

Abstract: lines 21-22: ' the conditions under which microbial processes cannot be neglected as organic carbon sinks in clouds' Please provide such information in the abstract.

We reworded this sentence as follows (l. 21/22):

*While also this estimate is very approximate, the analysis of the uncertainties and ranges of all parameters suggests **that high concentrations of metabolically active bacteria in clouds might represent an efficient sink for organics.***

Section 2.4.2: It is unclear which year of MODIS cloud data has been used. It would be nice to show the derived map of cloud volume and the ecosystem map that are later used to derive the numbers in Table 2 (in page 8 that is erroneously numbered as Table 1). What grid size is used for these calculations? How many grid points are used to derive the $F_{cloud}$ over each ecosystem type? This is an important Table for the budget estimates that are further presented in the manuscript; therefore, it has to be well documented.

We agree that our description of the MODIS data and cloud cover was too short (cf also our response to the comments by the other referee). We used data for spring (March – May) of MODIS Terra data, averaged over twelve years (2000 – 2011). We visually overlaid the map (Figure R-1) to the map by Burrows et al. (2009b), Figure R-2)

[Figure]

*Figure R-1: Seasonal mean daytime cloud fraction from Terra (2000-2011) for March – May, this figure is Figure 2b in King et al., (2013)*

[Figure]

*Figure R- 2: Lumped ecosystem classes, based on the Olson World Ecosystems (Olson et al., 1992); Figure 1 in Burrows et al., 2009b.*

This way, we estimated the cloud cover for the major ecosystems as listed by Burrows et al. (2009), based on the categories defined by Olson et al. (1992). We agree with the referee that a more detailed view could be used to characterize small scale features. However, given the conceptual nature of our study that builds upon the categories as used in the previous study by Burrows et al. (2009) for primary bacteria emissions, we think that our approach is sufficient to give (i) a reasonable estimate of cloudiness above the various ecosystems and (ii) sufficient detail of its concept to be refined in future studies.

Adding also a column with cloud fraction over each ecosystem as suggested by the other reviewer will be a significant improvement.

We extended Table 2 and added the cloud cover $F_{clc}$.

**Table 2:** Surface coverage of ecosystems on Earth surface (Burrows et al., 2009b), *approximate cloud coverage $F_{clc}$ above the ecosystems, estimated based on maps of annual cloud cover data obtained by MODIS,* and estimated time fraction bacteria spend in clouds ($F_{cloud}$)

| | % of Earth surface [a] | $F_{clc}$ | $F_{cloud}$ |
|---|---|---|---|
| All | 100 | 0.6 | 0.15 |
| Tundra | 3.3 | 0.4 | 0.1 |
| Grassland | 2.2 | 0.7 | 0.2 |
| Coastal | 0.2 | 0.4 | 0.1 |
| Wetlands | 0.6 | 0.5 | 0.15 |
| Crops | 3.0 | 0.7 | 0.3 |
| Land ice | 3.1 | 0.4 | 0.1 |
| Deserts | 3.7 | 0.2 | 0.05 |
| Forests | 7.0 | 0.9 | 0.25 |
| Shrubs | 5.8 | 0.3 | 0.1 |
| Seas | 71.0 | 0.7 | 0.2 |

[a] Data from Burrows et al. (2009b) and Olson *(1992)*

It is also unclear

1) how the value of 0.15 is derived for FCloud,

We modified the previous text as follows (l. 181ff):

In general, cloud contact times, i.e., the time air spends in a cloud, are dependent on cloud depth and vertical velocity (Feingold et al., 2013). This small-scale information is not consistently available for the large regions as covered by the ecosystems listed in **Table 1**.

 (King et al., 2013)  (Pruppacher and Jaenicke, 1995; Wang et al., 2000) In order to give an estimate of the cloud processing time over the various large ecosystems as identified by Olson et al. (1992), we use the approximate cloud fractions during spring averaged for 2000 – 2011 from MODIS Terra (e.g., Figure 2b in (King et al., 2013)). While this representation gives only some snapshot of cloudiness that varies over smaller spatial and temporal scales. However, given the conceptual nature of our study that builds upon the categories as used in the previous study by Burrows et al. (2009) for primary bacteria emissions, our approach seems sufficient to give (i) a reasonable estimate of cloudiness above the

various ecosystems and (ii) enough detail of its concept to be refined in future studies.

Globally, a range of cloud thicknesses of 1.4 – 1.9 km has been derived (Table 1 in (Wang et al., 2000)) from which we use h = 1.5 km as a single value for the average cloud thickness. Assuming further that globally > 90% of all liquid clouds reside in the lowest 6 km of the atmosphere (Δz = 6 km) (Pruppacher and Jaenicke, 1995), we can convert the cloud coverage as obtained from satellite data into cloud volume fractions using Eq-x:

$$F_{cloud} = F_{clc} \cdot \frac{cloud\ thickness\ [km]}{\Delta z\ [km]} = F_{clc} \cdot \frac{1.5\ km}{6\ km} \tag{3}$$

Comparison of previous estimates of global cloud coverage of 60% (Pruppacher and Jaenicke, 1995) and the volume fraction of liquid clouds within the atmosphere of 15% (Lelieveld and Crutzen, 1990) generally supports this relationship. The resulting $F_{cloud}$ values are summarized in **Table 2** together with the percentage area fraction of each ecosystem of the Earth surface, taken from Burrows et al. (2009b) and originally obtained from Olson (1992), and the cloud coverage data.

2) whether the category 'Seas' in Table 2 contains also the sea-ice.

No, the category 'land-ice' does not include sea-ice. The categories by Olson et al., do not include sea-ice as a separate category. There is no data available for bacteria cell concentrations above sea ice; however, we assume that cell concentrations and microbial activity are low there due to the lack of primary sources and because of low temperatures, respectively. We added this information to the text (l. 224ff):

According to the definition of the categories as suggested by Olson et al. (1992), the category 'land ice' does not include sea ice. It can be expected that above sea ice the sources and metabolic activity of bacteria are also very low (Martin et al., 2009) and thus can be likely neglected on a global scale.

Line 215: explain to what you refer when writing 'other formation rates

We reworded the text and replaced 'other formation rates' by 'formation rates in the individual ecosystems'. (l. 221)

'Caption of figure 3. The reference to Burrows et al is incomplete.

We completed the reference.

Line 273: 'sensitivity' I think 'uncertainty' is more appropriate here.

We agree. We replaced 'sensitivity' by 'uncertainty' (l. 285).

Table 3: maximum range for Ccell (1E9) can you comment who and where has measured this ?

**A:** As indicated as a comment in the Table, this value of $C_{cell}$ originates from the work by Paez-Rubio et al. (2005) and it is considered the highest expectable value for the concentration of airborne bacteria. This was observed above the surface of an agricultural field irrigated with nondisinfected effluent from a wastewater

storage lagoon in Mexico. Aerosol sampling was performed with biosamplers (impingers) and enumeration of total bacteria was done by epifluorescence microscopy observation. We added information about this particular location in Table 3 and in more detail in l. 253:

$C_{cell,max}$ was measured  above a wastewater storage lagoon and can be considered the highest expectable value of ambient bacteria (Paez-Rubio et al., 2005)

Table 3: comment for mcell first line: 'assuming that they' to avoid confusion replace 'they' by 'cells'.

As suggested, we clarified the sentence and replaced 'they' by 'cells'.

Lines 283-285: discussion about fungal spores: I do not see why this is discussed here. Please remove or rephrase sentence to better fit in the discussion

**A:** As the section starts out with discussing total primary biological particles, we think that mentioning fungal spores here is relevant as they contribute a major fraction to total PBA. We agree with the reviewer that the wording was not clear and thus we modified the text as follows (l. 294ff):

Bacteria usually comprise only a small mass fraction of total PBA; a major fraction is composed of fungal spores. Thus, their emissions are generally estimated to be larger in mass than those of bacteria.  their global emissions (25 Tg yr$^{-1}$) was suggested to contribute 23% to total primary organic aerosol (Heald and Spracklen, 2009). An estimate of fungal spore emissions based on tracer compounds resulted in predicted 50 Tg yr$^{-1}$ (Elbert et al., 2007)

Equation 6: Fc needs to be defined earlier, now it is defined in line 339.

We agree that the definition of Fc should have been included right after Equation (7). We moved the text to l. 348:

Equation 7 includes an additional factor $F_C$  that accounts for the microbial selectivity towards only some organics by each bacteria type (e.g. Šantl-Temkiv et al., 2013; Bianco et al., 2019).

Also check units in this equation.

We thank the referee for noticing the mistake in the equation. We corrected it as follows:

$$R_{WSOC,Bact}\left[\frac{g_C}{L(aq)s}\right] = \quad -F_{CO2}\ k_{Bact}\left[\frac{L(aq)}{cell\cdot s}\right]\cdot C_{Cell,aq}\left[\frac{cell}{L(aq)}\right]\cdot F_C\cdot C_{WSOC}\left[\frac{g_C}{L(aq)}\right]\cdot F_{cloud} \qquad (6)$$

Whereas the cell concentration in cloud water can be replaced by

$$C_{Cell,aq}\left[\frac{cell}{L(aq)}\right] = C_{cell,g}\left[\frac{cell}{m^3(g)}\right] / LWC\left[\frac{m^3(g)}{L(aq)}\right] \qquad (7)$$

with $C_{Cell,aq}$ and $C_{WSOC}$ being the concentrations of bacteria and water-soluble organic carbon in cloud water, respectively, and $C_{cell,g}$ ambient cell concentrations in the gas phase (e.g. Table 1).

Line 340-342: provide an uncertainty range for Fc instead of one value. How this uncertainty is affecting the here presented estimates of WSOC loss by bacteria?

To our knowledge, $F_c$ or any similar parameter has not been determined experimentally on ambient cloud water or aerosol samples. $F_c$ is the fraction of organic material that can be metabolized by bacteria in clouds (i.e. the potential of organic compounds to be biodegraded, which depend on their bioavailability and chemical formulae and structure). In general, only a small fraction of all cloud water organics (~15%) can be speciated on a molecular basis (Herckes et al., 2013). We added to the manuscript the text below (l. 354ff) as the justification of $F_C = 0.5$. As we discuss a range of $F_C$ (0.2 – 0.8) in Table 4, we refrained from adding a range here in the text.

Given the complexity of the organic matter in the atmosphere, the numerous organic molecules existing in cloud water and their variable susceptibility to biodegradation, this fraction is hard to specify with precision. Bianco et al. (2019) observed experimentally by FT-ICR-MS during laboratory incubation of cloud water that out of the 2178 compounds detected, 1094 were actually utilized by bacteria (~ 50%). Assuming that all these compounds were equally abundant, one could conclude that 50% of all cloud water organics were prone to be microbiologically consumed (i.e. $F_C = 0.5$). More quantitative support of this assumption could be given based on the fact that preferably small oxygenated organics are taken up by bacteria. Compilations of speciated cloud water organics have shown that small acids, such as formic and acetic acid, comprise a large fraction (up to ~30%) of the characterized fraction of cloud water organics (e.g. Figure 6, (Herckes et al., 2013). Lab experiments have shown that these acids are degraded by bacteria (Amato et al., 2007; Vaïtilingom et al., 2013).

Line 364: 'slightly higher contribution of chemical reactions to WSOC loss'. Figure 4 shows that for $Y_{voc}$ equal 0.5 the chemical loss can be double the bacteria loss. This is not 'slightly higher'. Please rephrase.

We rephrased the text as follows (l. 385ff):

The results in *Figure 4*  suggest that the microbial rates may be smaller than the chemical ones under most conditions. Overall, the values shown in Figure 4 only differ by a factor of ~2.5 which might imply that there are conditions under which chemical and biological processes in the aqueous phase compete.

**A:** Thank you for noticing this typo mistake. We modified to ~30 Tg/yr, consistently with Figure 4 (l. 396)

Line 377: the authors claim that the WSOC losses are smaller than the predicted production rates of SOAaq. However, when accounting for the range of these rates, there is no significant difference. Furthermore, this result will depend on the assumed Fc, so please rephrase.

We agree with the reviewer that our statements were not accurate. Part of this is due to the fact that aqSOA and WSOC are not necessarily comparable. There is clearly an overlap between these two groups of organics; however, they do not denote the same mass as WSOC might include volatile compounds that do not contribute to aqSOA after drop evaporation and not all aqSOA might be water soluble.

We added (l. 401ff):

For the parameters chosen in our estimate, they are also smaller than the predicted production rates of aqSOA in clouds of 13.1 - 46.8 Tg year-1 (Lin et al., 2014) or 20 - 30 Tg yr-1 (Liu et al., 2012). This estimate represents organic carbon sources and sinks in general. It should be noted that the organics included in WSOC and aqSOA, respectively, might not be identical.

Line 433: I think 0.7 Tg/yr should be 3.7 Tg/yr

We agree with the referee that the reference to the average value of 0.7 Tg/year is confusing. As we had stated before, we present here the value of 3.7 Tg/year as the 'best estimate' and replaced the number accordingly (l. 459)

Line 458: in our study provides.

Thanks for noticing this. We removed 'is' in the sentence.

In addition, please provide references for

1) Figure 1 in its caption

**A:** The schematic in Figure 1 is conceptual of the fact that bacteria either grow in size, divide, stay dormant or die. There is no specific and unique reference for this, so we included the following references: (Si et al., 2017)(Norris, 2015) for bacteria cells growing in size and dividing, (Kaprelyants et al., 1993; Price and Sowers, 2004) for cell dormancy, and (Engelberg-Kulka et al., 2006) for cell death.

We added in l. 67ff:

whereas bacteria dormancy and death do not lead to any change in cell mass (Kaprelyants and Kell, 1993; Price and Sowers, 2004)

2) the value of 7 to 14% in Line 159.

The values of 7-14% for BGE were taken from the reference cited in the sentence before (Eiler et al., 2003). In order to make it clearer, we connected the two sentences by (l. 158)

BGE for planktonic bacteria range from < 0.4% to 80% with the highest values for eutrophic conditions (Eiler et al., 2003). In turn, in the same study, it was shown that, when substrate availability is limited, values from ~7% to ~14% are generally observed.

**Reviewer #3**

This is a very interesting and important study that identifies and tackles a major gap in the aerosol cloud interactions, that is already lacking primary biological particles to a large extent and comes up with some rough estimates of the secondary biological particles. I find the paper suitable to be published in ACP, given that some issues raised below are answered.

We thank the referee for his/her positive evaluation of our manuscript. All comments are addressed in detail below.

1) How about sea-ice? Is it considered together with land-ice or not considered at all?

Sea ice is not considered in the major categories as defined by Olson et al. (1992) and as used in the global model study on primary bacteria emissions by Burrows et al. (2009). We added (l. 224ff):

According to the definition of the categories as suggested by Olson et al. (1992), the category 'land ice' does not include sea ice. It can be expected that above sea ice the sources and metabolic activity of bacteria are also very low (Martin et al., 2009) and thus can be likely neglected on a global scale.

2) Similarly, urban sources? Why are not hey represented as they can be a large source of bacteria due to human existence?

We focused on the large-scale ecosystems as suggested in the framework by Burrows et al. that, in turn, was based on the ecosystem categories defined by Olson. We agree with the referee that urban areas might be a particular source of bacteria and added the following text (l. 251ff):

However, spatial deviations might be present in particular locations, such as cell concentrations of ~$7 \cdot 10^5$ - $4 \cdot 10^6$ cells m$^{-3}$  that were found  during haze periods in China (Li et al., 2018; Xie et al., 2018), and even $10^9$ m$^{-3}$ above a wastewater storage lagoon which can be considered the highest expectable value of ambient bacteria (Paez-Rubio et al., 2005). Using the framework presented in the present study, SBA formation in such rather spatially limited areas can be estimated if growth rates of the individual bacteria types were available.

Generally, we'd like to note that this study aims at providing a first estimate of the contribution of bacteria to secondary biological aerosols (SBA), and providing a frame for future investigations in case where mode specific cases need to be studied. For now, this only includes the major ecosystems; given the general lack of data about bacteria abundance and activity in the atmosphere, and the large uncertainties associated with bacteria emissions (see Tables 3 and 4) from surfaces, we indeed intentionally simplified the global system. However, we completely agree that including more detailed information would be highly interesting and

needed, so we hope that our study will give in the model in the future and explore different scenarios of bacteria emission, cloud cover, temperature, etc.

3) Is it possible to provide with a formula that calculates $F_{cloud}$ based on ecosystem, corresponding cloud fraction from MODIS and the conversion factor in order to be able to reproduce the values in Table 2? Table 2 can be updated to include the cloud fraction over each ecosystem.

We agree that our description of the MODIS data and cloud cover was too short (cf also our response to the comments by the other referee). We used data for spring (March – May) of MODIS Terra data, averaged over twelve years (2000 – 2011). We visually overlaid the map (Figure R-1) to the map by Burrows et al. (2009b), Figure R-2)

[Figure]

***Figure R-3: Seasonal mean daytime cloud fraction from Terra (2000-2011) for March – May, this figure is Figure 2b in King et al., (2013)***

[Figure]

***Figure R- 4: Lumped ecosystem classes, based on the Olson World Ecosystems (Olson et al., 1992); Figure 1 in Burrows et al., 2009b.***

This way, we estimated the cloud cover for the major ecosystems as listed by Burrows et al. (2009), based on the categories defined by Olson et al. (1992). We agree with the referee that a more detailed view could be used to characterize small scale features. However, given the conceptual nature of our study that builds

upon the categories as used in the previous study by Burrows et al. (2009) for primary bacteria emissions, we think that our approach is sufficient to give (i) a reasonable estimate of cloudiness above the various ecosystems and (ii) sufficient detail of its concept to be refined in future studies. We modified the text as follows (l. 181ff):

In general, cloud contact times, i.e., the time air spends in a cloud, are dependent on cloud depth and vertical velocity (Feingold et al., 2013). This small-scale information is not consistently available for the large regions as covered by the ecosystems listed in **Table 1**.

 (King et al., 2013)  (Pruppacher and Jaenicke, 1995; Wang et al., 2000) In order to give an estimate of the cloud processing time over the various large ecosystems as identified by Olson et al. (1992), we use the approximate cloud fractions during spring averaged for 2000 – 2011 from MODIS Terra (e.g., Figure 2b in (King et al., 2013)). While this representation gives only some snapshot of cloudiness that varies over smaller spatial and temporal scales. However, given the conceptual nature of our study that builds upon the categories as used in the previous study by Burrows et al. (2009) for primary bacteria emissions, our approach seems sufficient to give (i) a reasonable estimate of cloudiness above the various ecosystems and (ii) enough detail of its concept to be refined in future studies.

Globally, a range of cloud thicknesses of 1.4 – 1.9 km has been derived (Table 1 in (Wang et al., 2000)) from which we use h = 1.5 km as a single value for the average cloud thickness. Assuming further that globally > 90% of all liquid clouds reside in the lowest 6 km of the atmosphere (Δz = 6 km) (Pruppacher and Jaenicke, 1995), we can convert the cloud coverage as obtained from satellite data into cloud volume fractions using Eq-x:

$$F_{cloud} = F_{clc} \cdot \frac{cloud\ thickness\ [km]}{\Delta z\ [km]} = F_{clc} \cdot \frac{1.5\ km}{6\ km}$$

Comparison of previous estimates of global cloud coverage of 60% (Pruppacher and Jaenicke, 1995) and the volume fraction of liquid clouds within the atmosphere of 15% (Lelieveld and Crutzen, 1990) generally supports this relationship. The resulting $F_{cloud}$ values are summarized in **Table 2** together with the percentage area fraction of each ecosystem of the Earth surface, taken from Burrows et al. (2009b) and originally obtained from Olson (1992), and the cloud coverage data.

4) Table 1 caption in section 2.4.2 should be corrected to Table 2.

We thank the referee for noticing this. We corrected the typo.

5) Is it possible to distinguish the different forest types or regions? It would be interesting to see these numbers above the amazons and boreal forests for example. Therefore, it would be interesting to show that global spatial distribution of this SBA source.

We fully agree that a finer spatial resolution of the individual ecosystems or subsystems would be interesting. However, the data sets of measured bacteria cells above different regions are rather limited. In the boreal forests near Hyytiälä, Finland, bacteria cell concentrations of $6323 \pm 13748$ m-3 were found (Helin et al., 2017), which is very close to the average concentration of 10000 cell m-3 shown in Table 1. Using an average cloud cover of 0.65 for boreal forests as derived by (Spracklen et al., 2008) translates into a cloud volume fraction of ~0.15 using the equation employed here. Thus, in a first approximation, it can be expected that the SBA source from boreal forests [ng m$^{-3}$] may be similar to the average one suggested here. Above the pristine forests of the Amazon, a rich biodiversity of prokaryotes has been found (Souza et al., 2019). However, in that study, the cell concentrations were not quantified. Given the large cloud cover of > 75% during the wet season in this area (Marquardt Collow et al., 2016), it can be expected that SBA formation might be significant there. Similarity, in the tropics, the great diversity of bacteria was characterized qualitatively (Gusareva et al., 2019); however, also in this latter study, no quantitative data on ambient cell concentration was given. In a study by (Huffman et al., 2012) a small mode of biological particles between 0.5 – 1 microm was identified during the AMAZE-08 campaign in the Amazon rain forest that likely contained bacteria. However, that study did not quantify them further. Preliminary data suggest concentrations of bacteria cells of $10^4 – 10^5$ m$^{-3}$ (personal communication, C. Pöhlker).

In order to highlight the lack of quantitative measurements of cell concentrations in the aforementioned regions and in general, we added the following text (l. 466ff):

The ecosystem categories in Table 2 represent fairly large regions. It might be expected that SBA formation rates are different on smaller spatial and/or temporal scales. For example, it has been shown that human activities in cities lead to high bacteria concentrations; also forests have been identified as significant sources of biogenic aerosol. However, detailed data on bacteria are sparse in such regions. While several recent studies have characterized the diversity of microorganisms in forested regions (rainforest, tropics) (Gusareva et al., 2019; Souza et al., 2019), these studies did not report cell concentrations which highlights the urgent need of additional measurements.

6) Line 236: 1% of the secondary aerosols.

We assume that the referee referred to line 296 ('Thus, SBA production can be estimated to be on the order of ~1% of the total aerosol sources'). We reworded as follows (l. 308):

Thus, SBA production can be estimated to be on the order of ~1% of the secondary aerosol sources.

7) Line 343: Where does the Fc=0.5 value come from, any reference or argument

To our knowledge, $F_c$ or any similar parameter has not been determined experimentally on ambient cloud water or aerosol samples. $F_c$ is the fraction of organic material that can be metabolized by bacteria in clouds (i.e. the potential of organic compounds to be biodegraded, which depend on their bioavailability and chemical formulae and structure). In general, only a small fraction of all cloud water organics (~15%) can be speciated on a molecular basis (Herckes et al., 2013). We added to the manuscript the text below as the justification of $F_C = 0.5$. As we discuss a range of FC (0.2 – 0.8) in Table 4, we refrained from adding a range here in the text. (l. 354ff):

[revised manuscript text omitted]

Marquardt Collow, A. B., Miller, M. A. and Trabachino, L. C.: Cloudiness over the Amazon rainforest: Meteorology and thermodynamics, J. Geophys. Res. Atmospheres, 121(13), 7990–8005, doi:10.1002/2016JD024848, 2016.

Martin, A., Hall, J. and Ryan, K.: Low Salinity and High-Level UV-B Radiation Reduce Single-Cell Activity in Antarctic Sea Ice Bacteria, Appl. Environ. Microbiol., 75(23), 7570, doi:10.1128/AEM.00829-09, 2009.

Norris, V.: Why do bacteria divide?, Front. Microbiol., 6, doi:10.3389/fmicb.2015.00322, 2015.

Olson, J.: World ecosystems (WE1.4): Digital raster data on a 10 minute geographic 1080 (2160 grid square), Global Ecosystem Database, Version 1, 1992.

Paez-Rubio, T., Viau, E., Romero-Hernandez, S. and Peccia, J.: Source Bioaerosol Concentration and rRNA Gene-Based Identification of Microorganisms Aerosolized at a Flood Irrigation Wastewater Reuse Site, Appl. Environ. Microbiol., 71(2), 804, doi:10.1128/AEM.71.2.804-810.2005, 2005.

Price, P. B. and Sowers, T.: Temperature dependence of metabolic rates for microbial growth, maintenance, and survival, Proc. Natl. Acad. Sci. U. S. A., 101(13), 4631–4636, doi:10.1073/pnas.0400522101, 2004.

Pruppacher, H. R. and Jaenicke, R.: The processing of water vapor and aerosols by atmospheric clouds, a global estimate, Atmos Res, 38(1–4), 283–295, doi:http://dx.doi.org/10.1016/0169-8095(94)00098-X, 1995.

Si, F., Li, D., Cox, S. E., Sauls, J. T., Azizi, O., Sou, C., Schwartz, A. B., Erickstad, M. J., Jun, Y., Li, X. and Jun, S.: Invariance of Initiation Mass and Predictability of Cell Size in Escherichia coli, Curr. Biol., 27(9), 1278–1287, doi:10.1016/j.cub.2017.03.022, 2017.

Souza, F. F. C., Rissi, D. V., Pedrosa, F. O., Souza, E. M., Baura, V. A., Monteiro, R. A., Balsanelli, E., Cruz, L. M., Souza, R. A. F., Andreae, M. O., Reis, R. A., Godoi, R. H. M. and Huergo, L. F.: Uncovering prokaryotic biodiversity within aerosols of the pristine Amazon forest, Sci. Total Environ., 688, 83–86, doi:10.1016/j.scitotenv.2019.06.218, 2019.

Spracklen, D. V., Bonn, B. and Carslaw, K. S.: Boreal forests, aerosols and the impacts on clouds and climate, Philos. Trans. R. Soc. Math. Phys. Eng. Sci., 366(1885), 4613–4626, doi:10.1098/rsta.2008.0201, 2008.

[revised manuscript text omitted]

---

## Author Response (AR2)

We thank the editor for her additional comments and the opportunity to further clarify our study. We addressed all comments in detail below. Editor comments are in blue, our responses in black and modified manuscript text in red. Line numbers refer to the revised manuscript without annotations.

We also realized that the Section numbering for Section 2.4 was wrong. We corrected it and replaced the Section numbers 2.4., 2.4.1, and 2.4.2. by 2.3. 2.3.1, and 2.3.2., respectively.

Comments to the Author:

I thank the authors for their point to point reply to the reviewers comments. The replies clarified significantly the procedure used for the global estimates provided. However, I consider that additional improvements are needed after which the paper can be published in ACP.

1- The authors now clarified that the MODIS cloud cover data are taken from the work of King et al 2013 – they precisely provide Figure 2b from that paper and R-1 in the replies. Thus I understand that the authors did not use the digital form of the data, since they do not provide a figure drawn by themselves. I also understand from their response that this figure is used to visually make the correspondence with the ecosystem map again taken from a publication in Burrows et al 2009b – here also no digital data seem to be used (figure R-2 provided in the replies). If this is the case, it has to be clearly stated. In such case, I also wonder how the mean cloud fraction over each ecosystem type that is provided in Table 2 is derived. For instance, forests over S America, where rainforests exist, based on these figures, are seen to correspond to cloud over of almost 1.0, while in Indonesia values as low as 0.4 can be seen from the figures. The methodology has to be clearly articulated in section 2.4.2. This section has been improved in clarity but does not state 1) that the correspondence between cloud fraction and ecosystems was done visually and 2) how the mean over the same ecosystem type with different cloud fractions is made. Furthermore, I consider that an estimate of the uncertainty in the numbers provided in Table 2 should be also provided.
2- 'Reasonable estimate' of cloudiness (line 197) is a vague statement. How much uncertainty tolerates 'reasonable'?

Author response: The editor is correct that we did not use digital data to derive the cloud fraction above the various ecosystems. As we consider our study a first order-of-magnitude estimate, we clarified this in Section 2.4.2:

```
In order to give an estimate of the cloud processing time over the various large
ecosystems as identified by Olson et al. (1992), we use the approximated derived
visually the cloud fractions during spring averaged for 2000 – 2011 from MODIS
Terra (e.g., based on Figure 2b by (King et al., 2013)). This representation
gives only a general view of cloudiness that varies over smaller spatial and
temporal scales. However, given the conceptual nature of our study that builds
upon the lumped ecosystem categories as used in the previous study by Burrows
et al. (2009) for primary bacteria emissions, our approach seems sufficient
appropriate to give (i) an reasonable order-of-magnitude estimate of cloudiness
above the various ecosystems and (ii) enough detail of its concept to be refined
in future studies on smaller spatial and temporal scales. This visual approach
to derive cloud fractions from the average maps neglects details on the
variability of cloud fractions among the same ecosystem category in different
geographic regions. For such categories (e.g., forests), we estimated a single
average value based on the surface-weighted fractions of the different regions.
```

3- Table 2 caption: MODIS cloud cover data from King et al (2013). Is it annual mean data as written in the caption or spring time as explained in the text?

Author response: These are the spring data. We clarified it in the captions as follows

```
Surface  coverage  of  ecosystems  on  Earth  surface  (Burrows  et  al.,  2009b),
approximate cloud coverage Fclc above the ecosystems, estimated based on maps of
annual cloud cover data obtained by MODIS Terra for spring (2000-2011), and
estimated time fraction bacteria spend in clouds (Fcloud)
```

4- Ecosystem category 'Seas' you mention that sea-ice is not included in this category, however from Figure R-2 we can see that 'Seas' cover also the North Pole region which is expected to have sea-ice. Therefore Sea-ice area seems to be included in the ecosystems as seas without differentiation. Please check again and revised lines 236-237 accordingly.

Author response: The original data base by Olson is has been developed only for land categories. In the study by Burrows et al. (Burrows et al., 2009), an additional category 'Seas' was introduced: *Seas: Oceans, Seas: Waters, including ocean and Inland Waters.*

None of the bacteria cell concentrations listed in Table 1 and discussed by Burrows et al. were measured above sea ice but at lower latitudes . We added to the text:

```
These   ecosystems   represent   lumped   categories   based   on   the   original
classification by Olson et al. (1992).The category 'seas' in Table 1 is not
included in the original categories as defined by Olson (1992). However, it was
added by Burrows et al (2009b) in order to represent a full global coverage.
There  are  only  a  few  studies  available  that  report  measurements  of  bacteria
numbers in the air above the Arctic. For example, report (1.3 ± 1.0) ·10³ cells
m⁻³ above partially glaciated surfaces in Southwest Greenland. Recent large scale
microbiological  studies  including  a  number  of  ground  based  stations  around  the
globe (Dommergue et al., 2019; Tignat-Perrier et al., 2019) reported bacteria
abundances  over  north  Greenland  in  the  Arctic  (Station  Nord)  3  to  4  orders  of
magnitude  lower  than  anywhere  elsewhere  on  the  planet  at  the  exception  of
Antarctica,  with  16S  rRNA  gene  copies  numbers  (representing  the  uppermost
expectable  cell  concentration)  of  (7.34  ±  9.22) ·10²  m⁻³.  In  this  area  the  air
content  is  affected  by  emissions  from  sea  ice,  the  Arctic  ocean  and  long-range
transport from northern Eurasia. This number is even lower than the one estimated
above  land  ice  by  Burrows  et  al.  (2009)(5000  m⁻³).  Thus,  it  can  be  assumed  that
the contribution of bacteria to SBA formation above sea ice is negligible.
```

5- Section 3.1.3. Although the flow of discussion has been improved, I still do not understand why for instance the authors discuss fungal spores and not pollen also, which is a large fraction of bioaerosols mass in the atmosphere.

Author response: We agree with the editor that pollen represent a major fraction of PBA in terms of mass but not in terms of particle number. The major difference between pollen and bacteria/viruses/fungal spores is the fact that the latter group might show metabolic processes, i.e., could possibly contribute to SBA mass whereas pollen will not.

We added to the text:

An estimate of aerosol emissions from the biosphere suggested a source strength of primary biological particles of 1000 Tg yr$^{-1}$. However, in this study, PBA was defined to include all cellular material, proteins, and their fragments. A global model study predicted total PBA emissions (bacteria, fungal spores and pollen) of 123 Tg yr$^{-1}$ of which bacteria comprised 0.79 Tg yr$^{-1}$, fungal spores 5.8 Tg yr$^{-1}$ and pollen 47 Tg yr$^{-1}$. These numbers  are similar to the ranges of 0.4 – 1.8 Tg bacteria yr$^{-1}$ and 84 Tg pollen yr$^{-1}$  (Burrows et al (2009b)), and 31 Tg fungal spores yr$^{-1}$. However, as pollen grains have usually sizes > 30 μm (Winiwarter et al., 2009), their atmospheric residence time is limited and thus their burden to total PBAP is relatively smaller than that of fungal spores (6.2 Gg vs 773.4 Gg, respectively) (Myriokefalitakis et al., 2017).

Author response: We modified/extended the text as follows:

We assume $0.2 \leq Y_{volC} \leq 0.5$, but in general, $YvolC$ depends on the WSOC composition, with higher values for more aged organics that are more readily oxidized to volatile products. This upper limit might be representative for fog water as characterized in Fresno (CA, USA) where about 50% of the dissolved organic carbon was composed of small acids (formic, acetic, oxalic) and aldehydes (formaldehyde, dicarbonyls). Small aldehydes are oxidized in the aqueous phase to the carboxylic acid; oxidation of carboxylic acids yields $CO_2$ (Ervens et al., 2003).

7- The authors want to make the difference between WSOC and aqSOA. However this distinction is not explained in the manuscript. I suggest that a sentence or two are added early in the document to explain to what WSOC and aqSOA differ according to the authors.

Author response: As we do not discuss aqSOA at all until this section as it is not relevant before, we did not follow the editor's suggestion to add text about earlier. WSOC and aqSOA certainly do not completely comprise the same compounds: While small carbon compounds (e.g., formaldehyde, acetaldehyde, formic, acetic acids) comprise a significant fraction of WSOC, e.g., (Herckes et al., 2013), they do not contribute to aqSOA as they evaporate during drop evaporation. In contrast, salt-forming dicarboxylates or oligomer compounds might not be water-soluble and thus do not contribute to WSOC. To our knowledge, there is no study to date that has quantified what fraction of (aq)SOA is WSOC and vice versa. We modified the text as follows in order to avoid the impression that WSOC and aqSOA might be considered comparable:

 We can conclude that the loss of WSOC by chemical and biological processes is relatively small (~10%) compared to the total removal of water-soluble organic carbon from the atmosphere by wet deposition derived from global models (293 Tg yr$^{-1}$ (Safieddine and Heald, 2017); 306 Tg yr$^{-1}$, (Kanakidou et al., 2012)).

Author response: We replaced year$^{-1}$ by yr$^{-1}$ everywhere in the manuscript.

9- Because WSOC is used for bacteria growth, it is not clear how much overall the aqSOA could change during bacteria growth and multiplications. However, bacteria may 'fix' very light organics, which are mainly dissolved in the water phase from the gas phase in the atmosphere, to the aqSOA, i.e. in forms that remain in the aerosol phase after evaporation. Thus I think the provided first estimate of SBA could be seen as an upper limit of impact on aqSOA.

Author response: We realized that we have to clarify better the different types of processes discussed in the study:

1) SBA formation is estimated based on cell generation rates, i.e. the cellular growth that leads to an increase in cell size and mass, and possibly cell number in the case of cell multiplication. The nutrients that are needed for cell generation do not only include WSOC, but also compounds containing nitrogen and phosphorus, essentially, along others. The elemental composition of bacteria cells varies widely depending on growth conditions. In cells growing with no nutrient limitation, carbon, nitrogen and phosphorus account for about 76%, 18% and 6% of the carbon, nitrogen and phosphorus masses, respectively. However, this can be drastically modified upon nutrient limitations. Accordingly, the cellular carbon content can vary from 33 to 241 fgC $\mu m^{-3}$, and the atomic C:N:P ratio from 52:8:1 to 163:25:1 (Chrzanowski and Kyle, 1996; Vrede et al., 2002). In our study, we assumed a total mass of $52 \cdot 10^{-15}$ g cell$^{-1}$, which thus comprises not only carbon but also the others elements constituting living organisms. Hence, the term $m_{cell}$, used for deriving SBA, is higher than the carbon mass that would be derived from WSOC uptake by cells. More parameterizations regarding cell mass, volume, and the proportion of carbon (or else) that it represents can be implemented later using the framework proposed here.

We added at the end of Section 2.2::

Bacteria cells are composed not only of carbon, but also other elements such as nitrogen, phosphorus, etc, the proportions of which can vary widely depending on nutrient condition (e.g., Vrede et al., 2002; Chrzanowski and Kyle, 1996). Hence, the total biological mass produced during cell growth and multiplication is higher than the amount of DOC incorporated.

2) Bacteria utilize WSOC for metabolic processes, i.e. for chemical reactions within the cells to sustain life. These organic substrates include both light organics dissolved form the gas phase (e.g. formic acid, formaldehyde) or CCN constituents (e.g. lactic acid, succinic acid). Under cloud conditions (i.e. oligotrophic conditions when substrate availability is limited) the major fraction of WSOC is converted into $CO_2$, thus, these light organics are not fixed by the bacteria but rather efficiently oxidized to volatile compounds.

We added at the beginning of Section 3.2.1:

These metabolic processes are typically enzyme-mediated chemical reactions within the bacteria cells that supply the necessary energy for the cells to maintain their viability. The cells typically utilize small organic compounds for these processes  leading to a decrease of water-soluble organic carbon (WSOC) mass within cloud droplets as bacteria convert these substrates into $CO_2$ (*Figure 1*).

The editor is correct that in analogy of primary biological aerosol (PBA) being a fraction of primary organic aerosol (POA), one may consider SBA as a fraction of aqSOA. However, as stated at the end of Section 3.1.3, overall SBA production likely only contributes a small fraction to total secondary aerosol and secondary organic aerosol sources. However, we would like to emphasize here once more that the purpose of our study is not to identify an additional major aerosol source in the atmosphere but to rather suggest a new source of biological aerosol mass that has very unique aerosol properties in terms of cloud formation (ice nucleation) and health effects. We think that this intention is already sufficiently clearly stated throughout the manuscript.

in the abstract: *…bacteria that have attracted a lot of attention due to their role in cloud formation and adverse health effects.*

in Section 3.1.3: The unique properties of biological aerosol material have been extensively discussed in the context of heterogeneous ice nucleation where it has been shown that even small amounts of biological material could have significant effects on clouds and precipitation (Möhler et al., 2008; Morris et al., 2004; Santl-Temkiv et al., 2015). Given the low ambient concentrations of ice nucleating particles and their high sensitivity to the ice/liquid partitioning in mixed-phase clouds (e.g., Ervens et al., 2011), a small change in biological mass possibly translates into significant changes in the evolution of cold clouds.

Conclusions: While these production rates make up ~1% of other major secondary aerosol formation rates (secondary organics or sulfate), their importance might differ on spatial or temporal scales. In addition, SBA production leads to an increase in biological aerosol mass which might sensitively affect physicochemical particle properties (e.g. ice nucleation ability).

10- Table 4, the provided range of Fc is 0.2 to less than 1 and not 0.8 as stated in the reply to the reviewer 3 comments. Please check maximum value. Also it would be nice to provide in the text a range of the estimate (or a % uncertainty) based on the min/max of all the parameters provided in Table 4. How such range will affect the conclusions of the manuscript?

Author response: Equation 7 is a linear equation, i.e. a change in any of the parameters included will change the result by the same factor. Thus, it can be easily seen that cell concentration and cell activity are the parameters with the largest range (two orders of magnitude) whereas the range of the other parameters (LWC, $F_{CO2}$, $F_c$) cover only a range of less than an order of magnitude. We do not think that it would be a meaningful estimate to assume situations where all minimum or maximum values are present at the same time, i.e. low (high) cell number concentration, low (high) cell activity, during low (high) cloudiness etc.

The numbers, we assumed represent the best estimate as they are based on average values reflecting average cell and cloud properties. On regional scales, these numbers can vary significantly; however, we do not deem a global estimate meaningful using the minimum or maximum values, respectively. In order to clarify this, we added at the beginning of Section 3.2.2:

[revised manuscript text omitted]

---

## Author Response (AR3)

Response to editor comments

We thank the editor Maria Kanakidou for her careful reading of our manuscript and guiding the review process. We revised the text as suggested below and also corrected a few typos etc in the final revised manuscript version.

Editor comment 1- In your reply to my earlier comment 1 you have added in page 8, lines 19-22: "This visual approach to derive cloud fractions from the average maps neglects details on the variability of cloud fractions among the same ecosystem category in different geographic regions. For such categories (e.g. forests), we estimated a single average value based on the surface-weighted fractions of the different regions."
How you can use 'visual approach' and also calculate 'surface-weighted fractions'? Some rephrasing is here needed.

Authors' response: This text was changed as follows (p. 6, l. 24ff)
*This visual approach to derive cloud fractions from the average maps neglects details on the variability of cloud fractions among the same ecosystem category in different geographic regions. For such categories (e.g. forests), the approximate surface contributions of the various regions were taken into account and averaged. fractions of the different regions.*

Editor comment 2- Page 14, lines 10-18 do not read well any more. To address my comment 5 you have added discussion in lines 10-14 that is now disconnected from what follows leading also to repetitions. I suggest you add all emission estimates for fungal spores (with appropriate discussion and references) together in line 11 and remove lines 15-18. Then start line 19 by: 'In addition none of the above mentioned fungal spore emission estimates includes…'

Authors' response: We reorganized this paragraph as follows (p. 9, l. 28ff)
*It was suggested that the global emissions of fungal spores (25 Tg yr$^{-1}$) comprise 23% of total primary organic aerosol (Heald and Spracklen, 2009). A study based on tracer compounds resulted in an emission estimate for fungal spores of 50 Tg yr$^{-1}$ (Elbert et al., 2007). None of these estimates include microbial activity as a source of biological mass. Our predicted SBA source of 3.7 Tg yr$^{-1}$ is restricted to the mass production by bacteria but is similar to predictions for primary bacteria emissions.*

Editor comment 3- Page 19,line 2: 'into the same difference' do you mean 'into a proportional change'?

Authors' response: Yes, that what we meant. We changed it accordingly (p. 13, l. 13/14):

*…a change in any of the parameters will translate into a proportional change in predicted WSOC loss.*